**Estimation of oceanic sub-surface mixing under a severe cyclonic storm**

**using a coupled atmosphere-ocean-wave model**

Kumar Ravi Prakash, Tanuja Nigam, Vimlesh Pant

Centre for Atmospheric Sciences, Indian Institute of Technology Delhi, New Delhi-110016

**Abstract**

A coupled atmosphere-ocean-wave model used to examine mixing in the upper oceanic layers under the influence of a very severe cyclonic storm Phailin over the Bay of Bengal (BoB) during 10-14 October 2013. The coupled model found to improve the sea surface temperature over the uncoupled model. Model simulations highlight prominent role of cyclone induced near-inertial oscillations in sub-surface mixing up to the thermocline depth. The inertial mixing introduced by the cyclone played a central role in deepening of the thermocline and mixed layer depth by 40 m and 15 m, respectively. For the first time over the BoB, a detailed analysis of inertial oscillation kinetic energy generation, propagation, and dissipation was carried out using an atmosphere-ocean-wave coupled model during a cyclone. A quantitative estimate of kinetic energy in the oceanic water column, its propagation and dissipation mechanisms were explained using the coupled atmosphere-ocean-wave model. The large shear generated by the inertial oscillations found to overcome the stratification and initiate mixing at the base of the mixed layer. Greater mixing was found at the depths where the eddy kinetic diffusivity was large. The baroclinic current, holding a larger fraction of kinetic energy than the barotropic current, weakened rapidly after the passage of the cyclone. The shear-induced by inertial oscillations found to decrease rapidly with increasing depth below the thermocline. The dampening of mixing process below the thermocline explained through the enhanced dissipation rate of turbulent kinetic energy upon approaching the thermocline layer. The wave-current interaction, non-linear wave-wave interaction were found to affect the process of downward mixing and cause the dissipation of inertial oscillations.

## 1. Introduction

The Bay of Bengal (BoB), a semi-enclosed basin in the northeastern Indian ocean, consists of surplus near-surface fresh water due to large precipitation and runoff from the major river systems of the Indian subcontinent (Varkey et al., 1996; Rao and Sivakumar, 2003; Pant et al., 2015). Presence of fresh water leads to salt-stratified upper ocean water column and formation of barrier layer (BL), a layer sandwiched between bottom of the mixed layer (ML) and top of the thermocline, in the BoB (Lukas and Lindstrom, 1991; Vinayachandran et al., 2002; Thadathil et al., 2007). The BL restricts entrainment of colder waters from thermocline region into the mixed layer thereby, maintains warmer ML and sea surface temperature (SST). The warmer SST together with higher tropical cyclone heat potential (TCHP) makes the BoB as one of the active regions for cyclogenesis (Suzana et al. 2007; Yanase et al. 2012, Vissa et al. 2013). Majority of tropical cyclones generate during the pre-monsoon (April-May) and post-monsoon (October-November) seasons (Alam et al., 2003; Longshore, 2008). The number of cyclones and their intensity is highly variable in seasonal and interannual time scales. The oceanic response to the tropical cyclone depends on the stratification of the ocean. The BL formation in the BoB is associated with the strong stratification due to the peak discharge from rivers in the post-monsoon season. The intensity of the cyclone largely depends on the degree of stratification (Neetu et al. 2012; Li et al. 2013). The coupled atmosphere-ocean model found to improve the intensity of cyclonic storm when compared to the uncoupled model over different oceanic regions (Warner et al., 2010; Zambon et al., 2014; Srinivas et al., 2016; Wu et al., 2016). Zambon et al. (2014) compared the simulations from the coupled atmosphere-ocean and uncoupled models and reported significant improvement in the intensity of storm in the coupled case as compared to the uncoupled case. The uncoupled atmospheric model produced large ocean-atmosphere enthalpy fluxes and stronger winds in the cyclone (Srinivas et al., 2016). When the atmospheric model WRF was allowed interactions with the ocean model, the SST found to be more realistic as compared to the stand-alone WRF (Warner et al., 2010; Gröger et al., 2015; Jeworek et al., 2017; Hagemann et al., 2017). Wu et al. (2016) demonstrated the advantage of using a coupled model over the uncoupled model in a better simulation of typhoon Megi's intensity.

Mixing in the water column has an important role in energy and material transference. Mixing in the ocean can be introduced by the different agents such as wind, current, tide, eddy,

and cyclone. Mixing due to tropical cyclones is mostly limited to the upper ocean, but the cyclone-
induced internal waves can affect the subsurface mixing. Several studies have observed that the
mixing in the upper oceanic layer is introduced due to the generation of near-inertial oscillations
(NIO) during the passage of tropical cyclones (Gonella, 1971; Shay et al., 1989; Johanston et al.,
2016). This mixing is responsible for deepening of ML and shoaling of the thermocline (Gill,
1984). The vertical mixing caused by storm-induced NIO has a significant impact on the upper
ocean variability (Price, 1981). The NIO are also found to be responsible for the decrement of SST
along the cyclone track (Chang and Anthes, 1979; Leipper, 1967; Shay et al., 1992; Shay et al.,
2000). This decrease in SST is caused by the entrainment of cool subsurface thermocline water in
the mixed layer into the immediate overlying layer of water. This cooling of surface water is one
of the reasons for the decay of cyclone (Cione and Uhlhorn, 2003). The magnitude of surface
cooling differs largely depending on the degree of stratification at the rightward to the cyclone
track (Jacob, 2003; Price et al., 1981).
The near-inertial process can be analyzed from the baroclinic component of currents. The
vertical shear of horizontal baroclinic velocities that is interrelated to buoyancy oscillations of
surface layers is utilized in various studies to have an adequate understanding of the mixing
associated with high-frequency oscillations, i.e. NIO (Zhang et al., 2014). The shear generated due
to NIO is an important factor for the intrusion of the cold thermocline water into the ML during
near-inertial scale mixing (Price et al., 1978; Shearman, 2005; Burchard and Rippeth, 2009). The
alternative upwelling and downwelling features of the temperature profile are an indication of the
inertial mixing. The kinetic energy bounded with these components of current shows a rise in
magnitude at the right side of cyclone track (Price et al., 1981; Sanfoard et al., 1987; Jacob, 2003).
The reason for this high magnitude of kinetic energy is linked with strong wind and rotating wind
vector condition of the storm. The spatial distribution of near-inertial energy is primarily controlled
by the boundary effect for inertial oscillations (Chen et al., 2017). The NIO is found to decline
with the decreasing depth and vanishes in the coastal regions (Schahinger, 1988; Chen et al., 2017).
The aim of this paper is to understand and quantify the near-inertial mixing due to the very
severe cyclonic storm Phailin in the BoB. Phailin was developed over the BoB in the northern
Indian ocean in October 2013. The landfall of Phailin occurred on 12 October 2013 around 15:30
GMT near Gopalpur district of Odisha state on the east coast of India. After the 1999 super
cyclonic event of the Odisha coast, Phailin was the second strongest cyclonic event that made
landfall on the east coast of India (Kumar and Nair, 2015). The low-pressure system developed in
the north of the Andaman Sea on $7^{th}$ October 2013, which transformed into a depression on $8^{th}$
October at 12 °N, 96 °E. This depression got converted to a cyclonic disturbance on $9^{th}$ October
and further intensified while moved to east-central BoB and opted the maximum wind speed of
200 km h$^{-1}$ at 03:00 GMT on $11^{th}$ October. Finally, landfall occurs at 17:00 GMT $12^{th}$ October.
More details on the development and propagation of Phailin can be found in the literature (IMD
Report, 2013; Mandal et al. 2015). The performance of the coupled atmosphere-ocean model in
simulating the oceanic parameters temperature, salinity, and currents during the Phailin is
discussed in Prakash and Pant (2017).
Most of the past studies on the oceanic mixing under cyclonic conditions were carried out
using in-situ measurements, which are constrained by the spatial and temporal availability. To the
best of our knowledge, the present study is first of its kind that utilizes a coupled atmosphere-
ocean-wave model over the BoB to estimate the cyclone-induced mixing and associated energy
propagation at the cyclone track and a location of maximum surface wind stress during the period
of peak intensity of the cyclone. The study also focuses on analyzing the subsurface distribution
of NIO with its vertical mixing potential. Further, the study quantifies the shear generated mixing
and the kinetic energy of the baroclinic mode of horizontal current varying in the vertical section
at a selected location during the active period of the cyclone. The dissipation rate of NIO and
turbulent eddy diffusivity are quantified.

**2. Data and Methodology**
**2.1 Model details**
Numerical simulations during the period of Phailin were carried out using the coupled
ocean-atmosphere-wave-sediment transport (COAWST), described in detail by Warner et al.
(2010). COAWST modeling system couples the three-dimensional oceanic model 'Regional
Ocean Modeling System' (ROMS), the atmospheric model 'Weather Research and Forecasting'
(WRF), and the wind wave generation and propagation model 'Simulating Waves Nearshore'
(SWAN). ROMS model used for the study is a free surface, primitive equation, sigma coordinate
model. ROMS is a hydrostatic ocean model that solves finite difference approximations of the
Reynolds averaged Navier-Stokes equations (Chassignet et al., 2000; Haidvogel et al., 2000,
Haidvogel et al., 2008; Shchepetkin and McWilliams, 2005). The atmospheric model component
in the COAWST is a non-hydrostatic, compressible model 'Advanced Research Weather Research
Forecast Model' (WRF-ARW), described in Skamarock et al. (2005). It has different schemes for
representation of boundary layer physics and physical parameterizations of sub-grid scale
processes. In the COAWST modeling system, appropriate modifications were made in the code of
atmospheric model component to provide an improved bottom roughness from the calculation of
the bottom stress over the ocean (Warner et al., 2010). Further, the momentum equation is modified
to improve the representation of surface waves. The modified equation needs the additional
information of wave energy dissipation, propagation direction, wave height, wavelength that are
obtained from wave component of the COAWST model. The spectral wave model SWAN, used in
the COAWST modeling system, is designed for shallow water. The wave action balance equation is
solved in the wave model for both spatial and spectral spaces (Booij et al. 1999). The SWAN model
used in the COAWST system includes the wave-wind generation, wave-breaking, wave-dissipation, and
nonlinear wave-current-wind interaction. The 'Model Coupling Toolkit' (MCT) used as a coupler in the
COAWST modeling system to couple different model components (Larson et al., 2004; Jacob et al., 2005).
The coupler utilizes a parallel-coupled approach to facilitate the transmission and transformation of various
distributed parameters among component models. MCT coupler exchanges prognostic variables from one
model to another model component as shown in Figure 1. The WRF model receives sea surface temperature
(SST) from the ROMS model and supplies the zonal (Uwind) and meridional (Vwind) components of 10-
m wind, atmospheric pressure (Patm), relative humidity (RH), cloud fraction (Cloud), precipitation (Rain),
shortwave (Swrad) and longwave (Lwrad) radiation to the ROMS model. The SWAN model receives
Uwind and Vwind from the WRF model and transfers significant wave height (Hwave) and mean
wavelength (Lmwave) to the WRF model. A large number of variables are exchanged between ROMS and
SWAN models. The ocean surface current components (Us, Vs), free surface elevations (η), and bathymetry
(Bath) provided to the SWAN from ROMS model. The wave parameters, i.e. Hwave, Lmwave, peak
wavelength (Lpwave), wave direction (Dwave), surface wave period (Tpsurf), bottom wave period
(Tmbott), percentage wave breaking (Qb), wave energy dissipation (DISSwcap), and bottom orbit velocity
(Ubot) provided from the SWAN to ROMS model through the MCT coupler. Further details on the
COAWST modeling system can be found in Warner et al. (2010).

**2.2 Model configuration and experiment design**

The coupled model was configured over the BoB to study Phailin during the period of 00 GMT 10 October – 00 GMT 15 October 2013. The setup of COAWST modeling system used in this study included fully coupled atmosphere-ocean-wave (ROMS+WRF+SWAN) models but the sediment transport is not included. A non-hydrostatic, fully compressible atmospheric model with a terrain-following vertical coordinate system, WRF-ARW (version 3.7.1) was used in the COAWST configuration. The WRF model used with 9 km horizontal grid resolution over the domain 65 °E-105 °E, 1°N-34 °N and 30 sigma levels in the vertical. The WRF was initialized with 'National Centre for Environmental Prediction' (NCEP) 'Final Analysis' (FNL) data (NCEPFNL, 2000) at 00 GMT 10 October 2013. The lateral boundary conditions in WRF were provided at 6-hour interval from the FNL data. We used the parameterization schemes for calculating boundary layer processes, precipitation processes, and surface radiation fluxes. The Monin-Obukhov scheme of surface roughness layer parameterization (Monin and Obukhov 1954) was activated in the model. The Rapid Radiation Transfer Model (RRTM) and cloud-interactive shortwave (SW) radiation scheme from Dudhia (1989) were used. The planetary boundary layer scheme YSU-PBL, described by Noh et al. (2003), was used. At each time step, the calculated value of exchange coefficients and surface fluxes off the land or ocean surface by the atmospheric and land surface layer models (NOAH) passed to the YSU PBL. The grid-scale precipitation processes were represented by WRF single-moment (WSM) six-class moisture microphysics scheme by Hong and Lim (2006). The sub-grid scale convection and cloud detrainment were taken care by Kain (2004) cumulus scheme.

A terrain following ocean model ROMS with 40 sigma levels in the vertical used in this study. The ROMS model domain used with zonal and meridional grid resolutions of 6 km and 4 km, respectively. This high resolution in ROMS enables to resolve mesoscale eddies in the ocean. The vertical stretching parameters, i.e. $\theta_s$ and $\theta_b$ were set at 7 and 2, respectively. The northern lateral boundary in ROMS was closed by the Indian subcontinent. The ROMS model observed open lateral boundaries in the west, east, and south in the present configuration. The initial and lateral open boundary conditions were derived from the 'Estimating the Circulation and Climate of the Ocean, Phase II' (ECCO2) data (Menemenlis et al., 2005). The ocean bathymetry was provided by the 2-minute gridded global relief (ETOPO2) data (National Geophysical Data Center, 2006). There was no relaxation provided to the model for any correction in the temperature, salinity, and current fields. The Generic-Length-Scale (GLS) vertical mixing scheme parameterized as the K-$\varepsilon$ model used (Warner et al., 2005). Tidal

boundary conditions were derived from the TPXO.7.2 (ftp://ftp.oce.orst.edu/dist/tides/Global)
data, which includes phase and amplitude of the M2, S2, N2, K2, K1, O1, P1, MF, MM, M4, MS4,
and MN4 tidal constituents along the east coast of India. The tidal input was interpolated from
TPXO.7.2 grid to ROMS computational grid. The Shchepetkin boundary condition (Shchepetkin,
2005) for the barotropic current was used at open lateral boundaries of the domain which allowed
the free propagation of astronomical tide and wind-generated currents. The domains of atmosphere
and ocean models are shown in Figure 2. The ROMS and SWAN were configured over the
common model domain shown with the shaded bathymetry data in Figure 2. The two locations
used for the time series analysis are marked with stars in Figure 2. These two locations, one on-
track and another off-track, were selected in the vicinity of the region of maximum surface cooling
and wind-stress during the passage of Phailin. The wave model SWAN was forced with the WRF
computed wind field. We used 24 frequency (0.04 - 1.0 Hz) and 36 directional bands in SWAN model. The
boundary conditions for SWAN were derived from the 'WaveWatch III' model. In the COAWST system,
the free surface elevations (ELV) and current (CUR) simulated by ocean model ROMS are provided to the
wave model SWAN. The Kirby and Chen (1998) formulation was used for the computation of currents.
The surface wind applied to the SWAN model (provided by WRF) used in the Komen et al. (1984) closure
model to transfer energy from the wind to the wave field. The baroclinic time step used in ROMS model
was 5 s. The SWAN and WRF models used with time steps of 120 s and 60 s, respectively. The coupled
modeling system allows the exchange of prognostic variables among the atmosphere, ocean, and
wave models at every 600 s. The SST simulation at high spatial and temporal resolutions enables
accurate heat fluxes at the air-sea interface and exchange of heat between the oceanic mixed layer
and atmospheric boundary layer. The surface roughness parameter calculated in the WRF model
based on Taylor and Yelland (2001), which involved parameters from the wave model.

**2.3. Methodology**

The baroclinic current component was calculated by subtracting the barotropic component

from the mean current with a resolution of 2 m in the vertical. The power spectrum analysis was
performed on the zonal and meridional baroclinic currents along the depth section of the selected
locations by using periodogram method (Auger and Flandrin, 1995). The continuous wavelet
transform using Morlet wavelet method (Lilly and Olhede, 2012) carried out to analyze the
temporal variability of the baroclinic current at a particular level of 14 m. The near-inertial
baroclinic velocities were filtered by the Butterworth $2^{nd}$ order scheme for the cutoff frequency
range of 0.028 to 0.038 cycle hr$^{-1}$. The filtered zonal ($u_f$) and meridional ($v_f$) inertial baroclinic
currents were used to calculate the inertial baroclinic kinetic energy ($E_f$) in $m^2 s^{-2}$ and inertial shear
($S_f$) following Zhang et al. (2014) using equation (1).

$$S_f^2 = (\frac{\partial u_f}{\partial z})^2 + (\frac{\partial v_f}{\partial z})^2 \qquad (1)$$

As the stratification is a measure of oceanic stability, the buoyancy frequency (N) was calculated
using equation (2)

$$N^2 = -\frac{g}{\rho}\frac{\partial \rho}{\partial z} \qquad (2)$$

Where $\rho$ is the density of seawater and g is the acceleration due to gravity.
The analysis of generation of the inertial oscillations and their dissipation was performed
on the basis of turbulent dissipation rate ($\epsilon$) and turbulent eddy diffusivity ($k_\rho$). These parameters
were calculated by using following formula (Mackinnon and Gregg, 2005; van der Lee and
Umlauf, 2011; Palmer et al., 2008; Osborn, 1980)

$$\varepsilon = \varepsilon_0 \left(\frac{N}{N_0}\right)\left(\frac{S_{lf}}{S_0}\right) \qquad (3)$$

$$k_\rho = 0.2 \, x \left(\frac{\varepsilon}{N^2}\right) \qquad (4)$$

Where $S_{lf}$ is the low shear background velocity, Values of $N_0 = S_0 = 3$ cycle per hour and $\varepsilon_0 =$
$10^{-8}$ W kg$^{-1}$.

**3. Results and Discussion**
**3.1. Validation of coupled model simulations**
The WRF model simulated track of Phailin was validated against the India Meteorological
Department (IMD) reported best-track of the cyclone. A comparison of the model-simulated track
with the IMD track is shown in Figure 3. Solid circles marked on both the tracks represent the 3-
hourly positions of the cyclone's center, as identified by the minimum surface pressure. The daily
positions of the centre of Phailin are labelled with the date. WRF model in the coupled
configuration does a fairly good job in simulating the track, translational speed, and landfall
location of Phailin. The positional track error was about 40 km when compared to IMD track of
Phailin. The stand-alone WRF model (not shown here) was found to simulate Phailin track almost
similar to the WRF in the coupled configuration. However, the intensity (surface wind speed) in
WRF stand-alone model was higher as compared to the coupled model. Figure 4 shows the
comparison of stand-alone and coupled WRF model simulated mean sea level pressure (MSLP),
wind speed, and wind direction at a buoy (BD09) location (marked with a blue circle in Figure 3).
It can be inferred from the figure that stand-alone WRF simulated a larger pressure drop and higher
wind speed as compared to buoy measurements. In addition to the cyclone-induced pressure drop
during 10-12 October, the semidiurnal variations in MSLP were observed in the buoy
measurements. These semidiurnal variations in MSLP, primarily due to the radiational forcing
(Pugh, 1987), were not captured by the model over the cyclone-influenced region. The WRF in
coupled model configuration shows better performance in simulating the surface wind speed and
pressure during Phailin. The exchange of wave parameters with the WRF model in coupled
configuration provides realistic sea surface roughness that resulted in improvement of surface wind
speed.

The SST simulated by the ROMS model in coupled and stand-alone configurations was

validated against the Advanced Very High Resolution Radiometer (AVHRR) satellite data on each
day for the period of Phailin passage over the BoB. The stand-alone WRF simulated parameters
were used to provide surface boundary conditions in the stand-alone ROMS model. Figure 5 shows
that the coupled model captures the SST spatial pattern reasonably well with about -0.5℃ bias in
northwestern BoB on 13-14 October. This order of bias in SST could be resulted from the errors
in initial and boundary conditions provided to the model.  The maximum cooling of the sea surface
observed on 13[th] October in the northwestern BoB in both, coupled model and observations. This
post-cyclone cooling primarily associated with the cyclone-induced upwelling resulting from the
surface divergence driven by the Ekman transport. Thus, the coupled model is reproducing
dynamical processes and vertical velocities reasonably well. The stand-alone ROMS model
overestimates the cyclone-induced cooling with -2.2 ℃ bias in SST on 13-14 October (Figure 5).
The stronger surface winds in stand-alone WRF cause the larger cold bias in stand-alone ROMS
model.

## 3.2. Cyclone-induced mixing

The coupled atmosphere-ocean-wave simulation is an ideal tool to understand air-sea
exchange of fluxes and their effects on the oceanic water column. Surface wind sets up currents
on the surface as well as initiate mixing in the interior of the upper ocean. In order to examine the
strength of mixing due to Phailin, the model simulated vertical temperature profile together with
the surface wind speed, zonal and meridional components of current, and kinetic energy at the on-
track and off-track locations are plotted in Figure 6. Comparatively stronger zonal and meridional
currents observed at the off-track location than the on-track location on 12 October. The larger
kinetic energy available at the off-track location leads to greater mixing resulting into a deeper
mixed layer on 12 October as compared to the on-track location. The surface wind speed at the on-
track location shows a typical temporal variation of a passing cyclone. The wind speed peaks,
drops, and attains second peak as the cyclone approaches, crosses over, and depart the location.
The surface currents forced by these large variations in wind speed and direction at the on-track
location results into comparatively weaker magnitude than the off-shore location.
The thermocline, defined as the depth of maximum temperature gradient, is usually
referred to a location dependent isotherm depth (Kessler, 1990; Wang et al., 2000). Over the BoB
region, the depth of 23ºC isotherm (D23) found to be an appropriate representative depth of the
thermocline (Girishkumar et al., 2013). Based on the density criteria, we calculated the oceanic
mixed layer depth (MLD) as the depth where density increased by 0.125 kg m$^{-3}$ from its surface
value. The inertial mixing introduced by the cyclone play central role in deepening of D23 and
MLD on 12$^{th}$ October 2013. The warmer near-surface waters mixed downward when the cyclone
crossed over this location. After the passage of cyclone, shoaling of D23 and MLD observed as a
consequence of cyclone induced upwelling that entrain colder waters from the thermocline into
the mixed layer. The temperature of the upper surface water (25 -30 m) decreased by 3.5°C from
its maximum value of 28 °C after the landfall of the cyclone on  12-13$^{th}$ October at the off-shore
location (Figure 6g). In response to the strong cyclonic winds, the D23 deepening by 40 m (from
50 m to 90 m) was observed during 04-12 GMT on 12 October. At the same time, the MLD,

denoted by a thick black line in Figure 6g, deepens by about 15 m. On the other hand, the on-track location showed cooling at the surface only for a short time on 13 October and the deepening of D23 and MLD were 20 m and 10 m, respectively. To examine the role of cyclone induced mixing in modulating the thermohaline structure of upper ocean, we carried out further analysis on the coupled model simulations as discussed in the following sections.

### 3.2.1. Kinetic energy distribution

During the initial phase of Phailin, the zonal and meridional currents were primarily westward and southward, respectively (Figures 6c, 6d, 6h, and 6i). However, on and after $12^{th}$ October when cyclone attains peak intensity and crosses over the location, alternative temporal sequences of westward/eastward in zonal current and southward/northward in meridional current were noticed in current profiles (Figure 6). The frequency of these reversals in zonal and meridional currents are recognized as near-inertial frequency generated from the storm at these locations. The direction and magnitude of currents represent a variability that corresponds to the presence of near-inertial oscillations at the selected locations. The kinetic energy (KE) of currents at various depths is a proxy of energy available in the water column that becomes conducive to turbulent and inertial mixing. Time series of KE associated with the barotropic and depth-averaged baroclinic components of current at the two point locations are illustrated in Figure 6e (on-track) and 6j (off-track). The KE associated with the baroclinic component found to be much higher than the barotropic component of current at both on-track and off-shore locations. The depth-averaged baroclinic and barotropic current components' KE also depict the impinging oscillatory behavior. The peak magnitude of KE in baroclinic and barotropic currents at the off-shore location found to be 1.2 $m^2$ $s^{-2}$ and $0.3 \times 10^{-2}$ $m^2$ $s^{-2}$, respectively on $12^{th}$ October at 08:00 GMT. Whereas the magnitude of KE in baroclinic and barotropic currents at the on-shore location was smaller than the off-shore location during the peak intensity of cyclone. The peak magnitude of kinetic energy in baroclinic current at the off-track location was more than double to that of on-track location. The comparatively smaller magnitude of KE at the on-shore location could be associated with the rapid variations in wind speed and direction leading to the complex interaction between subsurface currents in the central region of the cyclone. It is worth noting that the time of peak KE in baroclinic currents coincide with the deepening of MLD and D23. Therefore, the KE generated in NIO is

responsible for sub-surface mixing that acts to deepen the mixed layer. The analysis suggests that
energy available for mixing process in the water column was mostly confined to the baroclinic
currents at various depths.

### 3.2.2. Primary frequency and depth of mixing

The power spectrum analysis was performed on the time series profiles at the two selected
locations - to get a distribution of all frequencies operating in the mixing process during the passage
of Phailin.  The power spectrum analysis performed on the zonal and meridional components of
the baroclinic current profile and shown in the Figure 7. It is clear from the figure that the tidal
(M2, the semidiurnal component of tide) and near-inertial oscillations (f) are the two dominant
frequencies on the surface during the cyclone Phailin. Under the influence of cyclonic winds, the
NIO signal was stronger (0.84 m2s-2) at the off-track than the on-track location. The depth
penetration of NIO was up to 50 m and 35 m at the off-track and on-track location, respectively.
The tidal frequency (M2) and inertial frequency (f) bands shown in the Figure 7 implies that the
inertial oscillations were dominant over the tidal constituent in zonal and meridional baroclinic
currents. At the off-track location, the largest power of the NIO was noticed at 14 m depth, but the
tidal oscillations were almost absent in the vertical section of baroclinic current (Figure 7). This
finding motivated us to analyze the significance and distribution of this sub-surface variability that
resulted in an anomalous deepening of MLD. The highest power of this signal at the off-track
location was associated within 0-15 m with the magnitude of 0.84 $m^2\,s^{-1}$ in zonal baroclinic current
and within 0-38 m with the magnitude of 0.76 $m^2\,s^{-1}$ in the meridional baroclinic current. These
signals, however, weaken with increasing depth and almost disappeared around 120 m depth.
These NIO were the strongest signals at the 14 m depth in the presence of local wind stress that
dominated the mixing compared to any other process. Other processes include the background
flows, the presence of eddies, variations in sea surface height, non-linear wave-wave and wave-
current interactions (Guan et al., 2014; Park and Watts, 2005).
The second order butterworth filter was applied to the baroclinic current components to get
the strength of NIO in the frequency range of 0.028 to 0.038 cycles $h^{-1}$ at the selected locations.
The filtered baroclinic current was further utilized to calculate the filtered inertial baroclinic KE
($E_f$ in $m^2s^{-2}$). The daily profiles of baroclinic KE were analysed at the two selected locations and
shown in Figure 8. The peak baroclinic KE differs from 0.14 $m^2s^{-2}$ at the on-track to 0.23 $m^2s^{-2}$ at
the off-track location on 12 October. As shown in Figures 6 and 7, the filtered baroclinic KE
profiles (Figure 8) confirm the dominant presence of NIO at the off-track location as compared to
the on-track location. The decay of NIO with the increasing depth was noticed at both the locations.
However, the NIO baroclinic KE penetrated up to 80 m in case of off-track as compared to only
50 m at the on-track location. The analysis, therefore, suggests that the NIO generated during the
Phailin were more energetic at the selected off-track location, which was also the location of
maximum surface cooling as noticed in Figure 5. Therefore, the further analysis in the subsequent
sections is limited to the off-track location only. To analyze the time distribution of the strong
NIO, wavelet transform analysis applied on the zonal and meridional baroclinic currents at 14 m
depth. The Scalogram, shown in Figure 9, depicts the generation of NIO signal at the off-track
location on 12[th] October that subsequently got strengthen and attains its peak value on the mid of
13[th] October. The energy percentage of the meridional component was always lower than the zonal
component. The peak values of energy percentage were found in the time periods between 1-1.3
days.

**3.2.3. Role of downward propagation of energy**

To investigate the energy propagation from the surface to the interior layers of upper-
ocean, we derived the rotary spectra (Gonella, 1972; Hayashi, 1979) of near-inertial wave numbers
and shown in Figure10. The daily averaged vertical wave-number rotary spectra provides a clear
picture of wind energy distribution in the sub-surface water. The anticyclonic spectrum ($A_m$) is
dominating over the cyclonic spectra ($C_m$) for the entire duration of the cyclone. This feature
indicates that the energy is propagating downward generated by these inertial oscillations. The
magnitude of these oscillations increased from initial stage up to 12[th] October and remained at high
energy density for the rest of the cyclone period. This downward directed energy initiated a process
of mixing between the mixed layer and the thermocline. This energy helps to deepen the mixed
layer against oceanic stratification by introducing a strong shear. The buoyancy of stratified ocean
was overcome to some extent by the shear generated that assist in mixing process during the very
severe cyclone. Alford and Gregg (2001) highlighted that in most of the cases, the energy of inertial
oscillations potentially penetrates the mixed layer but suddenly drops down as it touches the
thermocline. The energy dissipation mechanism studied in few other studies (Chant, 2001; Jacob,
2003).The 2-layer model described by Burchard and Rippeth (2009) illustrated the process of
generation of sufficient shear to start mixing near the thermocline. Their simple model ignored the
effect of the lateral density gradient, mixing, and advection. Burchard et al. (2009) mentioned four
important parameters for the shear generation, i.e. surface wind stress ($P_S S^2$), bed stress ($-D_b S^2$),
interfacial stress ($-D_I S^2$), and barotropic flow ($P_m S^2$). Utilizing simulations from our coupled
atmosphere-ocean-wave model, we calculated individual terms as suggested by Burchard et al.
(2009) and presented in Figure 10.  Surface wind stress found to be the most dominating term in
modulating the magnitude of bulk shear during the stormy event. Rest of the terms were relatively
weaker and, therefore, contributing only marginally to the variability of the bulk shear.
To examine the generation and dissipation of these inertial oscillations, the shear generated
by the near-inertial baroclinic current ($S_f^2$) and turbulent kinetic energy dissipation rate ($\varepsilon$) were
calculated and analyzed. The shear produced by inertial oscillations increased at 20-80 m depth
and higher magnitude was associated with peak wind speed of cyclone (Figure 12a). This shear
overcome the stratification (Figure 12b), represented by buoyancy frequency $N^2$, and played
important role in mixing and deepening of the thermocline and mixed layer on 12[th] October.  The
value of kinetic energy dissipation rate ($\varepsilon$) increased from $4 \times 10^{-14}$ to $2.5 \times 10^{-13}$ W kg$^{-1}$ on
approaching the thermocline (Figure 12c). The increase in $\varepsilon$ indicates the weakening of the shear
generated by the inertial waves leading to the fast disappearance of these baroclinic instabilities
from the region. The non-linear interaction between the NIO and internal tides together with the
prevailing background currents cause rapid dissipation of kinetic energy in the thermocline. Guan
et al. (2014) also reported an accelerated dampening of NIO associated with the wave-wave
interactions between NIO and internal tides. The background currents found to modify the
propagation of NIO (Park and Watts, 2005). The magnitude of the turbulent eddy diffusivity ($K_\rho$),
shown in Figure 12d, implies that the greater mixing takes place within the mixed layer where
$K_\rho$ was high ($6.3 \times 10^{-11}$ to $1.2 \times 10^{-11}$ m$^2$ s$^{-1}$). The daily averaged values of $\varepsilon$ and $K_\rho$ were $1.2 \times 10^{-13}$
W kg$^{-1}$ and $1.5 \times 10^{-10}$ m$^2$ s$^{-1}$, respectively on 12[th] October, which were higher as compared to
the initial two days of the cyclonic event. Results from the present study, as well as the conclusions
from the past studies, indicate that wave-current interaction, mesoscale processes, and wave-wave
interaction can affect the process of downward mixing and cause the dissipation of inertial
oscillations.

## 4. Conclusions

Processes controlling the sub-surface mixing were evaluated under the high wind speed regime of the severe cyclonic storm Phailin over the BoB. A coupled atmosphere-ocean-wave (WRF+ROMS+SWAN) model as part of the COAWST modeling system was used to simulate atmospheric and oceanic conditions during the passage of Phailin cyclone. A detailed analysis of model-simulated data revealed interesting features of generation, propagation, and dissipation of kinetic energy in the upper oceanic water column. Deepening of the MLD and thermocline by 15 m and 40 m, respectively were explained through the strong shear generated by the inertial oscillations that helped to overcome the stratification and initiate mixing at the base of the mixed layer. However, there was a rapid dissipation of the shear with increasing depth below the thermocline. The peak magnitude of kinetic energy in baroclinic and barotropic currents found to be 1.2 $m^2$ $s^{-2}$ and 0.3×10$^{-2}$ $m^2$ $s^{-2}$, respectively. The power spectrum analysis suggested a dominant frequency operative in sub-surface mixing that was associated with near-inertial oscillations. The peak strength of 0.84 $m^2$ $s^{-1}$ in the zonal baroclinic current found at 14 m depth at a location in northwestern BoB. The baroclinic kinetic energy remained higher (> 0.03 $m^2$ $s^{-2}$) during 11-12 October and decreased rapidly after that. The wave-number rotary spectra identified the downward propagation, from the surface up to the thermocline, of energy generated by inertial oscillations. A quantitative analysis of shear generated by the near-inertial baroclinic current showed higher shear generation at 20-80 m depth during peak surface winds. Analysis highlights that greater mixing within the mixed layer takes place where the eddy kinetic diffusivity was high (> 6×10$^{-11}$ $m^2$ $s^{-1}$). The turbulent kinetic energy dissipation rate increased from 4×10$^{-14}$ to 2.5×10$^{-13}$ W kg$^{-1}$ on approaching the thermocline that dampened mixing process further down into the thermocline layer. The wave-current interaction, mesoscale processes, and wave-wave interaction increased the dissipation rate of shear and, thereby, limited the downward mixing up to the thermocline. The coupled model found to be a useful tool to investigate air-sea interaction, kinetic energy propagation, and mixing in the upper-ocean. The results from this study highlight the importance of atmosphere-ocean coupling for a better understanding of oceanic response under the strong wind conditions. The proper representation of kinetic energy propagation and oceanic mixing have

applications in improving the intensity prediction of a cyclone, storm surge forecasting, and
biological productivity.

**Author contribution:** KRP and TN performed model simulations and analyzed data. VP prepared
the manuscript with contributions from all co-authors.

**Acknowledgements**
ECCO2 is a contribution to the NASA Modeling, Analysis, and Prediction (MAP)
program. The study benefitted from the funding support from Ministry of Earth Sciences, Govt. of
India and Space Applications Centre, Indian Space Research Organisation. High Performance
Computing (HPC) facility provided by IIT Delhi and Department of Science and Technology
(DST-FIST 2014 at CAS), Govt. of India are thankfully acknowledged. Authors are thankful to
Dr. Lingling Xie for his productive suggestions. Graphics generated in this manuscript using Ferret
and NCL. The constructive comments from three anonymous reviewers helped to improve the
manuscript. TN and KRP acknowledge MoES and UGC-CSIR, respectively for their PhD
fellowship support.

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

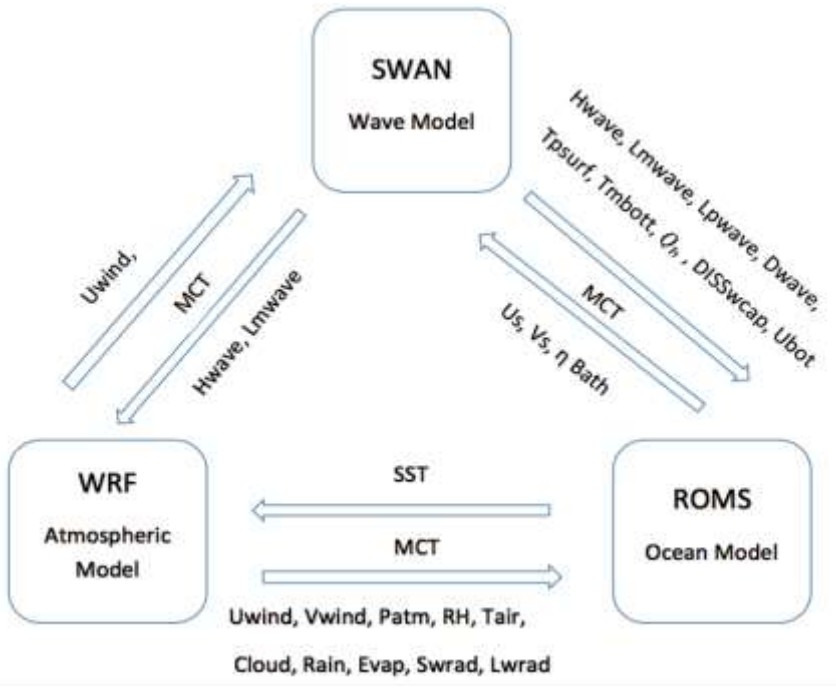


**Figure 1:-The block diagram showing the component models WRF, ROMS, and SWAN of the COAWST modeling system together with the variables exchanged among the models. MCT- the model coupling toolkit is a model coupler used in the COAWST system.**




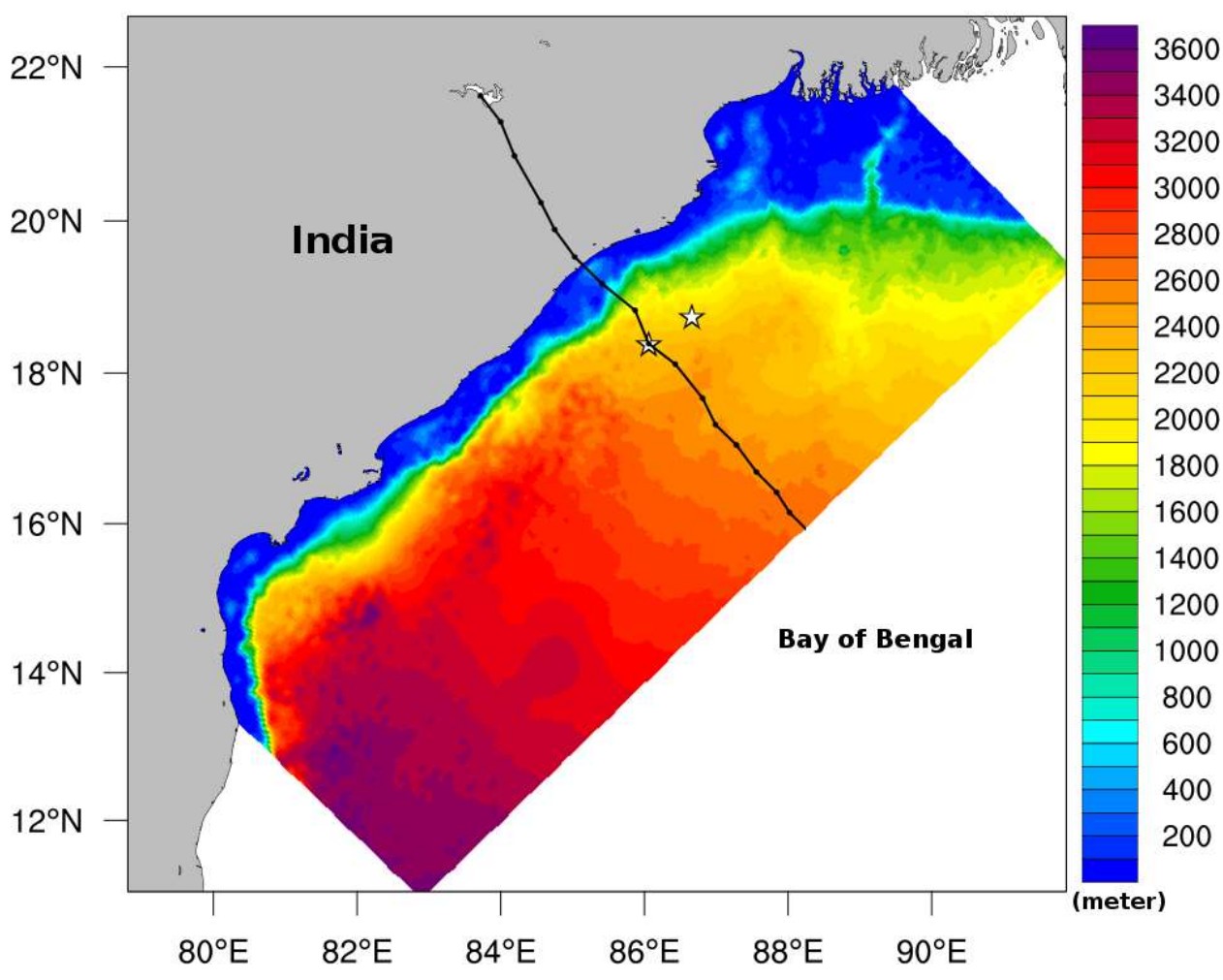

**Figure 2:-COAWST model domain (65°-105 °E, 1°-34 °N) overlaid with GEBCO bathymetry (m). Locations used for time-series analysis are marked with stars.**

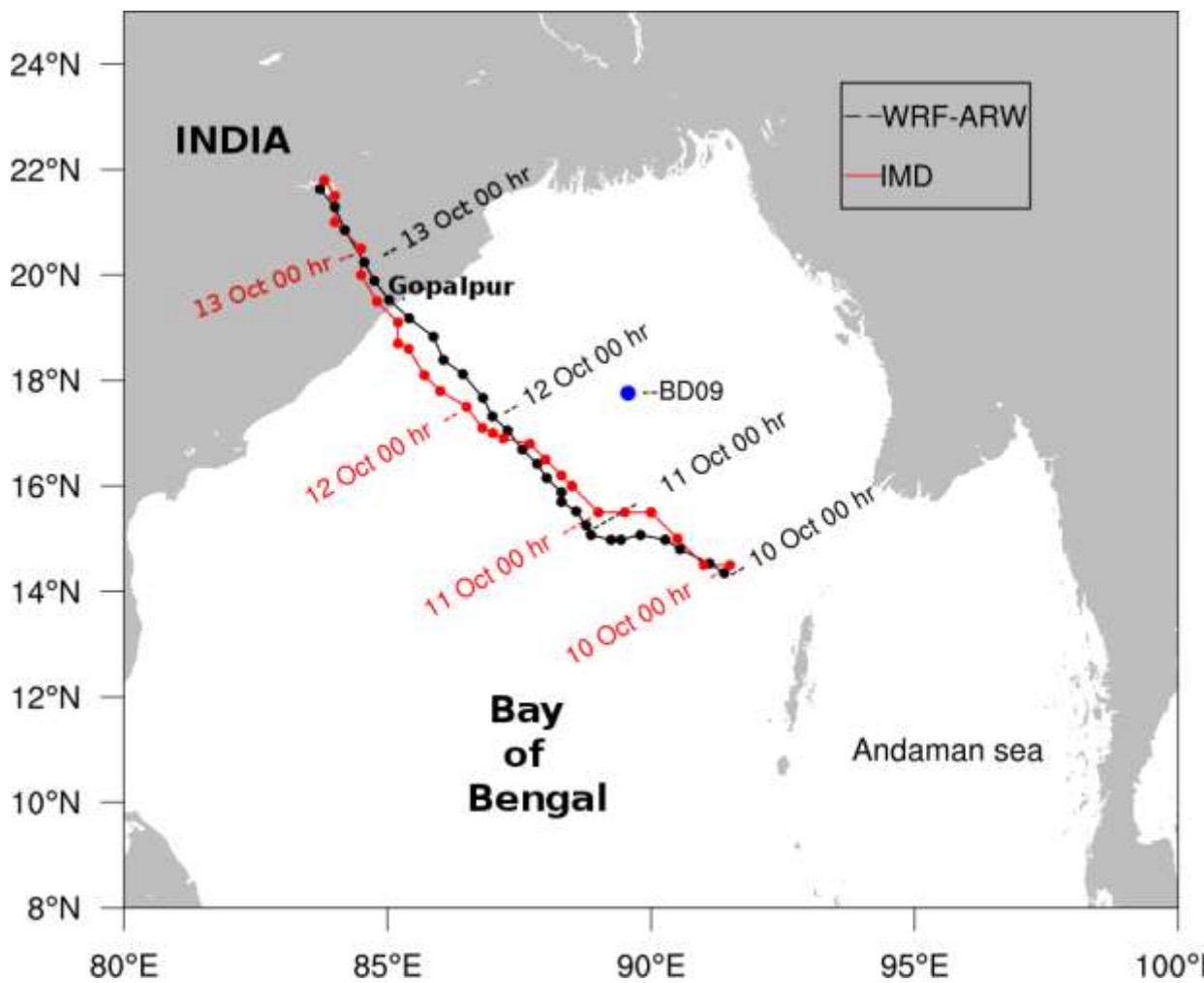


**Figure 3:- Tracks of Phailin simulated by the coupled model (black) and IMD reported (red). The**
**3-hourly positions of the center of Phailin marked with solid circles and the daily position at 00 hr**
**are labelled with the dates. Location of buoy BD09 is marked with a blue circle.**



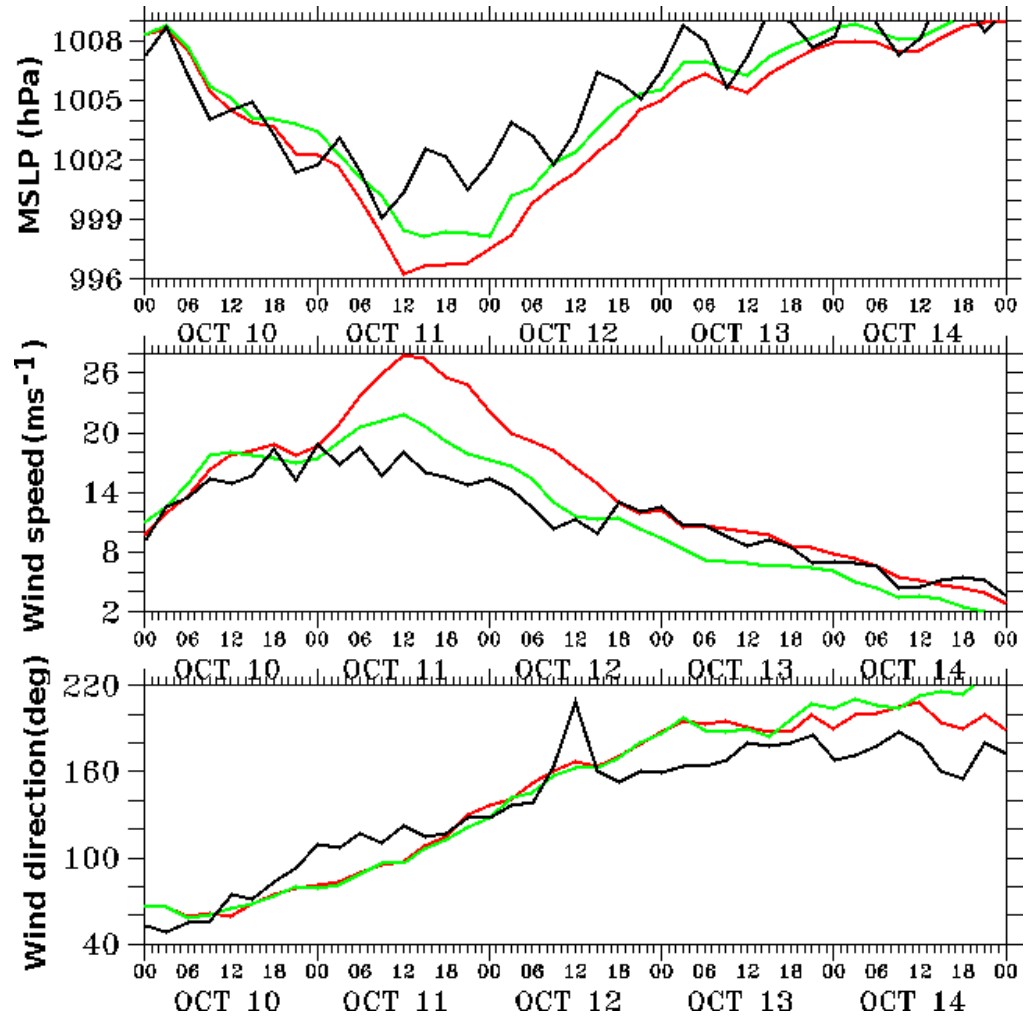


Figure 4: Comparison of coupled model (green), stand-alone WRF model (red), and observations from a buoy BD09 (black) for the (top panel) mean sea level pressure (hPa), (middle panel) wind speed (ms$^{-1}$), and (bottom panel) wind direction (degree).



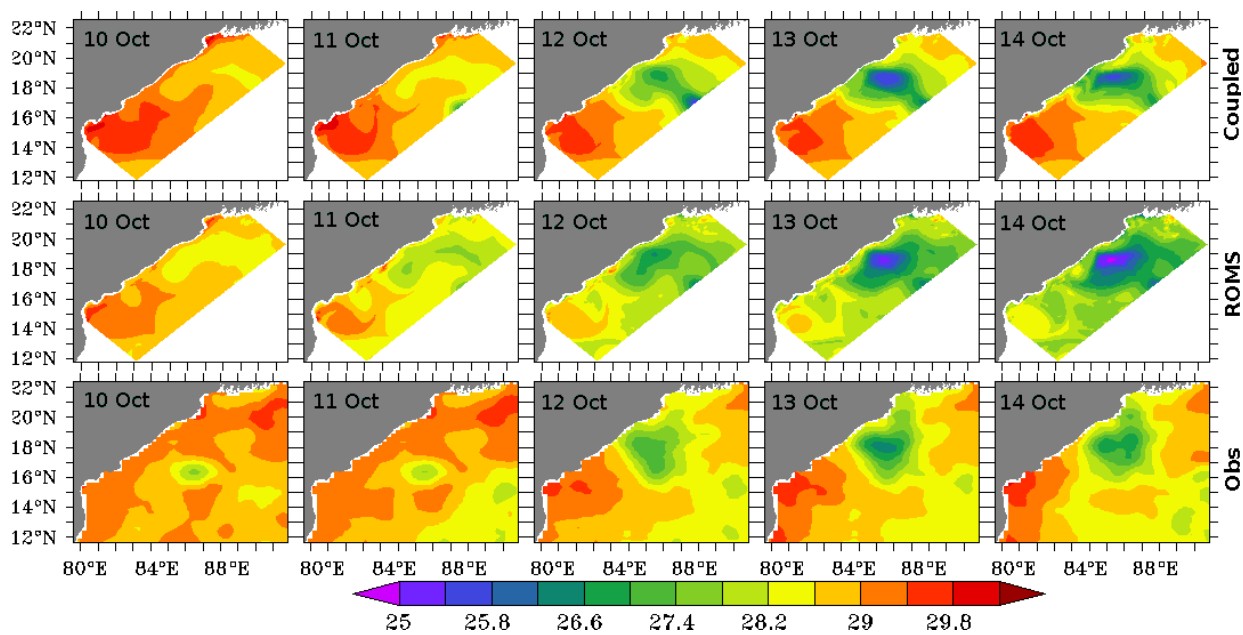


**Figure 5:- The daily averaged sea surface temperature (SST) in °C simulated by the coupled model (upper panel), stand alone ROMS model (middle panel), and observed from AVHRR sensor on the satellite (lower panel)..**



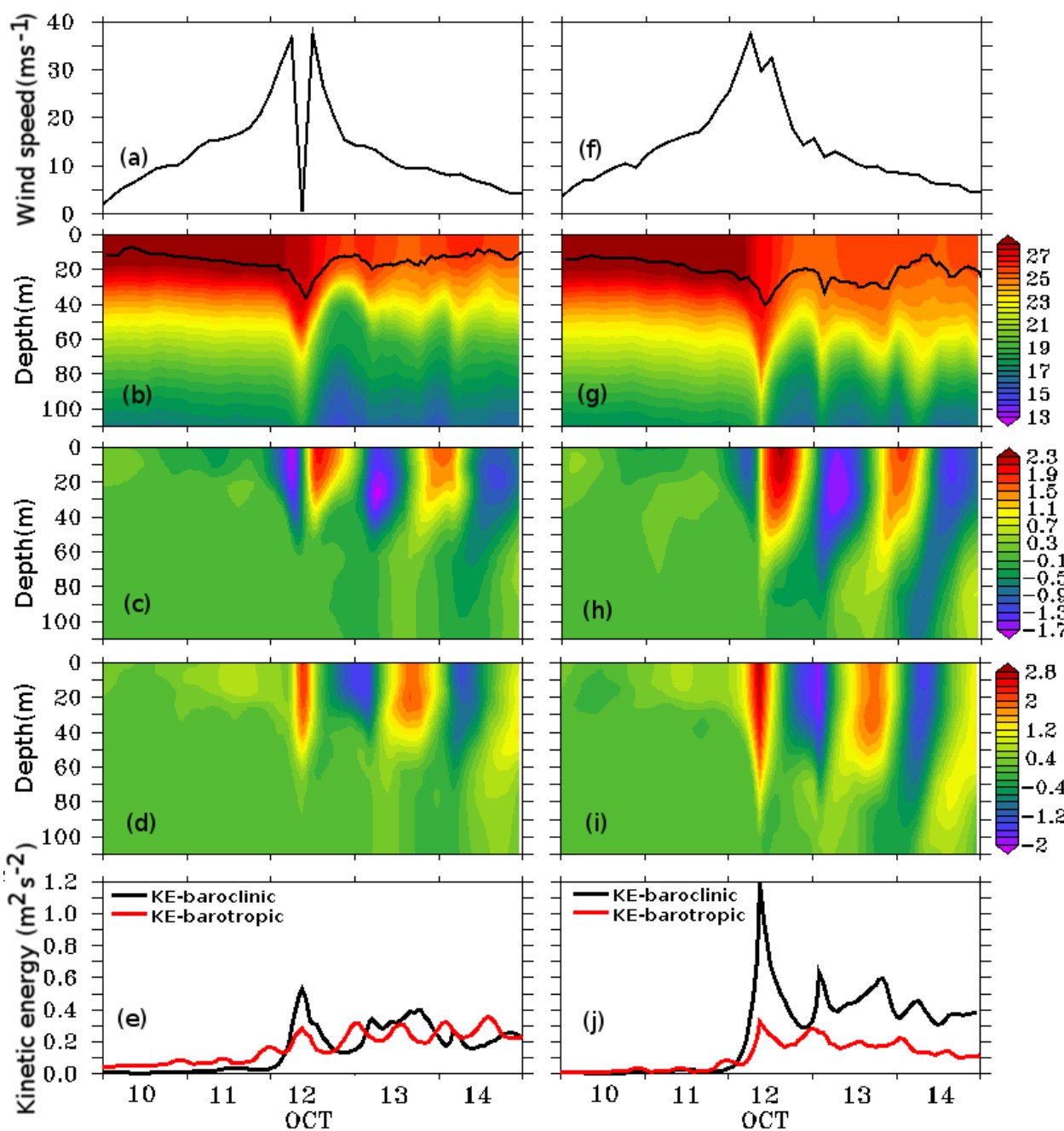


**Figure 6: Coupled model simulated and diagnosed variables at the on-track (left panel) and off-track (right panel) locations. (a, f) Surface wind speed (ms$^{-1}$), (b, g) temperature profile (ºC) and mixed layer depth (black line), (c, h) u-component of current (ms$^{-1}$), (d, i) v-componenet of current (ms$^{-1}$), (e, j) Kinetic energy of baroclinic (m$^2$s$^{-2}$) and barotropic (×10$^{-2}$ m$^2$s$^{-2}$) current.**





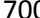

Figure 7:- The power spectrum analysis ($m^2s^{-1}$) performed on the simulation period at the on-track (upper panel) and off-track (lower panel) locations for (a, c) baroclinic zonal current and (b, d) baroclinic meridional current.

705

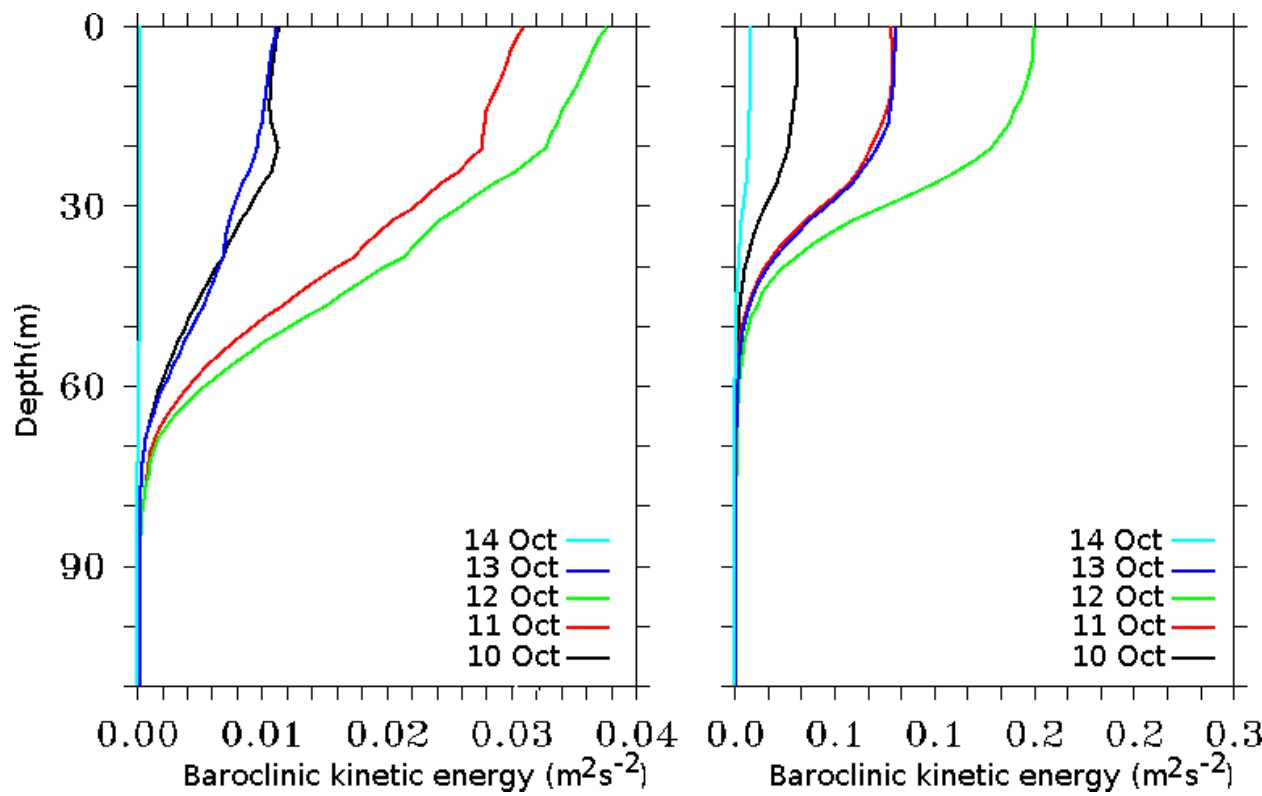

Figure 8: Daily averaged baroclinic kinetic energy (m²s⁻²) profile at the on-track (left) and off-track (right) locations as marked with stars in Figure 2.

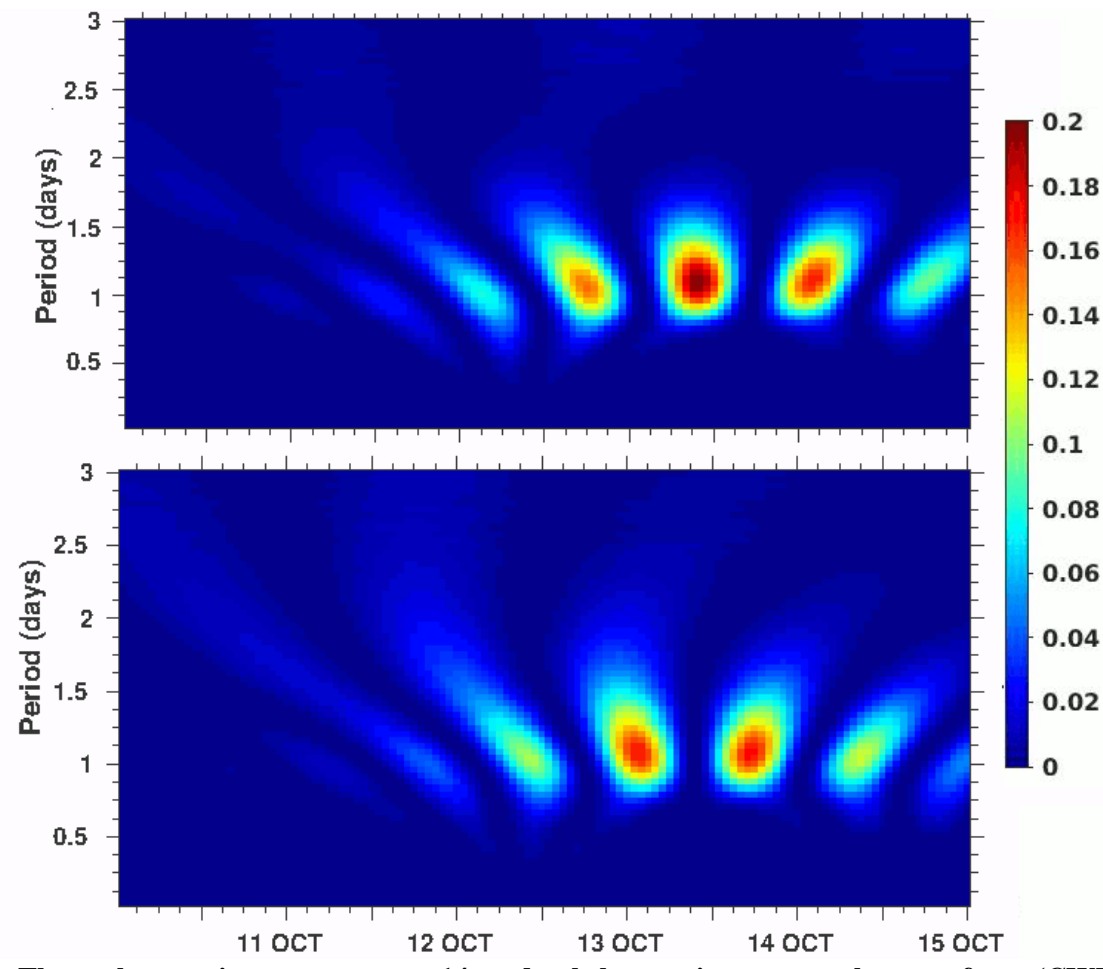

**Figure 9:- The scalogram in percentage at 14 m depth by continuous wavelet transform (CWT) method. Wavelet scalogram shown for the zonal baroclinic current (upper panel) and for the meridional baroclinic current (lower panel).**

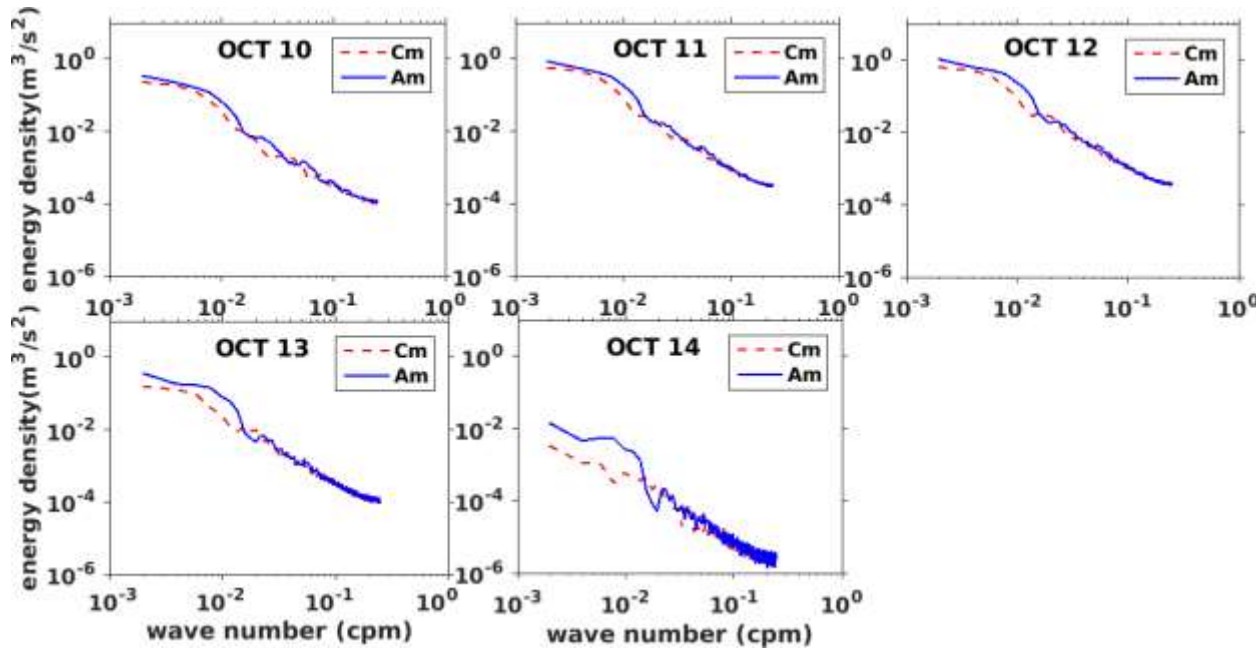

722

**Figure 10:- The daily averaged vertical wave-number rotary spectra of near inertial oscillations. The anticyclonic and cyclonic spectra are represented in blue and dotted red lines respectively.**

725

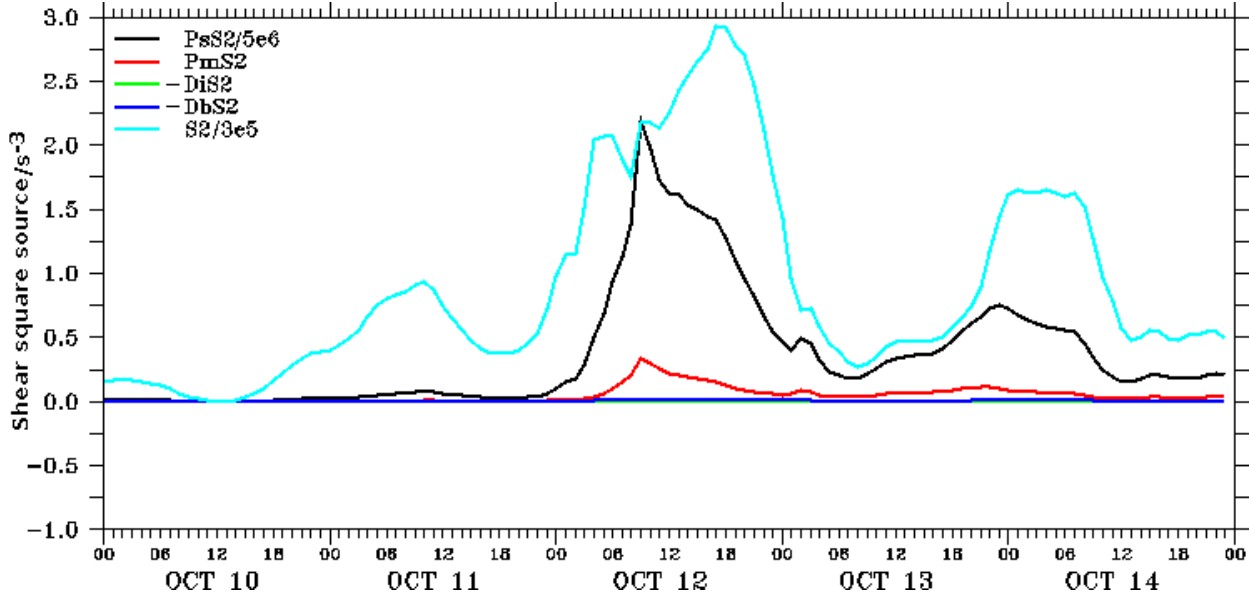

726

**Figure 11:- The model simulated bulk properties at the selected point location. The vertical shear square axis is multiplied with a factor of $10^{-6}$. The magnitude of bulk shear squared $S^2$ (cyan color), surface wind stress $P_sS^2$ (black color), barotropic effect $P_mS^2$ (red color), bottom stress $-D_bS^2$ (blue color), interfacial friction $-D_iS^2$ (green color)**




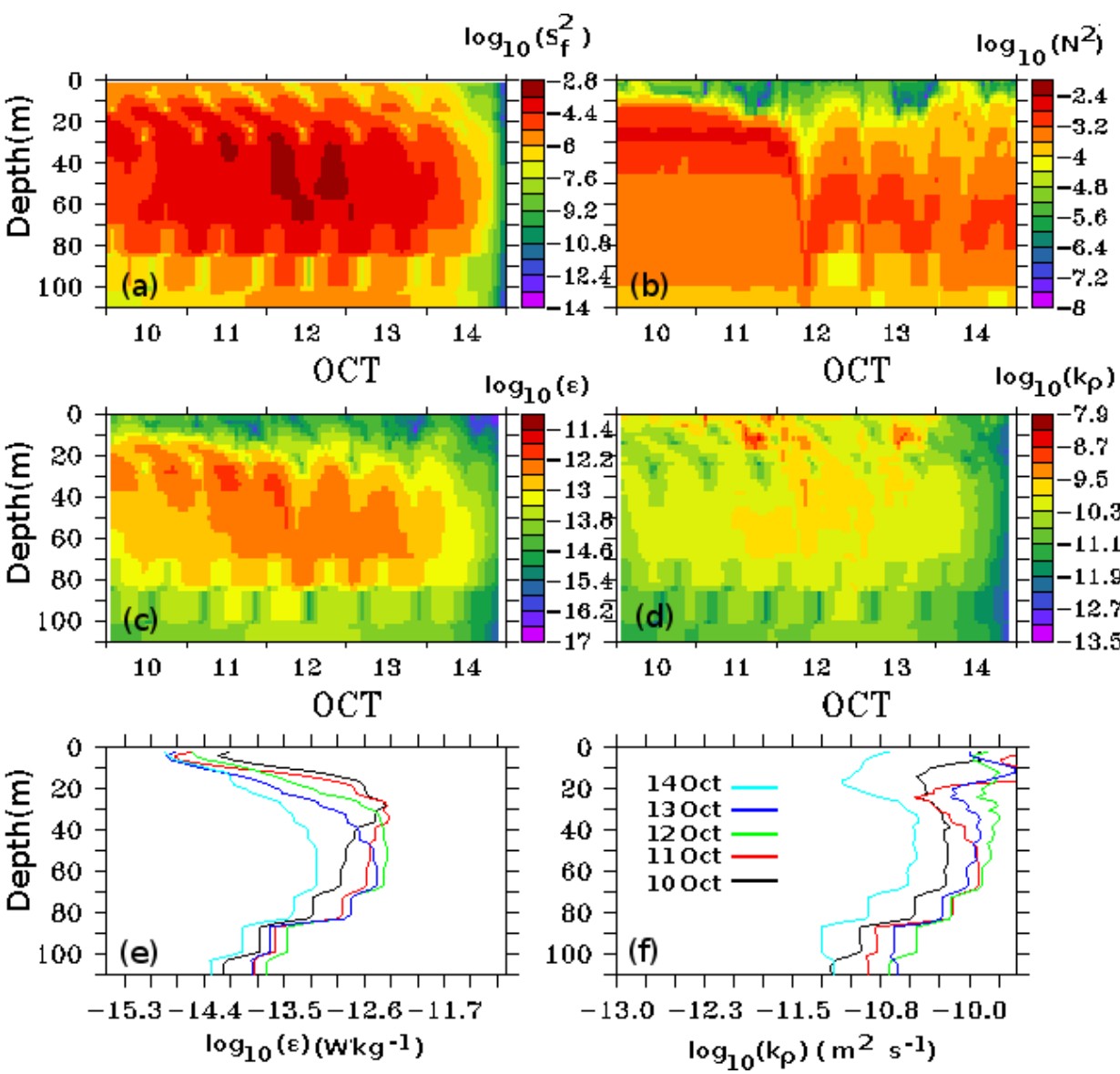

**Figure 12:- Profiles of (a) velocity shear log$_{10}$(S$^2$), (b) buoyancy frequency log$_{10}$(N$^2$), (c) turbulent kinetic energy dissipation rate log$_{10}$ ($\varepsilon$), (d) turbulent eddy diffusivity log$_{10}$ (K$\rho$), (e) and (f) are daily averaged turbulent kinetic energy dissipation rate and turbulent eddy diffusivity respectively**
