# Peer review of "Estimation of oceanic sub-surface mixing under a severe cyclonic storm"

_Ocean Science, 2017_

## Referee Comment (RC1) · Anonymous Referee #1 · 17 Nov 2017

Comments on the paper "Estimation of oceanic sub-surface mixing under a severe cyclonic storm using a coupled atmosphere-ocean-wave model"

General comments The authors investigate the effects of sub-surface mixing in the ocean under severe storm conditions. The introduction and the reference give the impression that the authors know very well the relevant publication and the overview they give is very nice. To my understanding the novel approach of the article is the use of a coupled atmosphere-ocean-wave model to investigate to simulate the atmospheric and oceanic properties on a very fine scale. The focus is on the generation, propagation and dissipation of kinetic energy in the ocean. I would definitely recommend the

publication of the article, although the English is not very well. Almost each sentence is missing an article or the third person "s" is neglected and Ocean is often spelled with capital O. This is not acceptable. In detail: The abstract is much longer than the conclusions which should be the other way around. And there is not substantial note to the model system used in the article although this is a very important point. Without the model, the investigation could not have happened. So the abstract should focus more on the novel approach and the details of the findings should be discussed in the conclusions. Page 3, line 65: NIO is one of the important factors, what are the others? Page 8, line 211: where are the 15m to be seen? The link between the description and the figures is not really strong. Page 9, line240ff: Unclear that the tidal and near/inertial oscillations are the two dominant frequencies. Line 250/251: sentence not understandable. What are the other processes?

---

## Author Comment (AC1) · 25 Nov 2017

Interactive reply to Anonymous Referee #1 of manuscript OS-2017-83 "Estimation of oceanic sub-surface mixing under a severe cyclonic storm using a coupled atmosphere-ocean-wave model" by Kumar Ravi Prakash et al.

General comments (1) The authors investigate the effects of sub-surface mixing in the ocean under severe storm conditions. The introduction and the reference give the impression that the authors know very well the relevant publication and the overview they give is very nice. To

my understanding the novel approach of the article is the use of a coupled atmosphere-ocean-wave model to investigate to simulate the atmospheric and oceanic properties on a very fine scale. The focus is on the generation, propagation and dissipation of kinetic energy in the ocean. I would definitely recommend the publication of the article, although the English is not very well. Almost each sentence is missing an article or the third person "s" is neglected and Ocean is often spelled with capital O. This is not acceptable.

Response: We sincerely thank the Referee for finding it publishable and providing constructive comments that helped to improve the manuscript. We thoroughly checked the manuscript for any missing articles, grammatical mistakes and made necessary corrections. 'Ocean' is now corrected as 'ocean'.

(2) In detail: The abstract is much longer than the conclusions which should be the other way around. And there is not substantial note to the model system used in the article although this is a very important point. Without the model, the investigation could not have happened. So the abstract should focus more on the novel approach and the details of the findings should be discussed in the conclusions.

Response: As suggested by the Referee, we made substantial changes in the Abstract and Conclusions. The abstract is shortened and detailed findings are now discussed in the Conclusions. The model details and model configurations are provided with more details in Section 2 of the revised version. We added a block diagram (Figure 1 in the revised manuscript) to clearly show the exchange of variables between component models WRF, ROMS, and SWAN in the COAWST modeling system. Discussion on variables exchanged added in the manuscript.

(3) Page 3, line 65: NIO is one of the important factors, what are the others?

Response: This sentence is now modified to clear any confusion. The NIO and surface wind stress can generate near-inertial scale mixing at the base of the mixed layer. Other processes such as the nonlinear interaction of NIO and internal tides, and background flows in the ocean can influence the NIO propagation and kinetic energy and affect the mixing process. Effects of other processes are mentioned at appropriate places (sections 3.3.2 and 3.3.3) in the revised manuscript.

(4) Page 8, line 211: where are the 15m to be seen? The link between the description and the figures is not really strong.

Response: We regret this mistake. The mixed layer depth (MLD) was calculated using the density criteria. We have now marked the position of MLD with a thick black line in Figure 4a (Figure 5a in the revised version). Description of figures elaborated to make the link between figures and text, and the flow between sections strong. (5) Page 9, line240ff: Unclear that the tidal and near/inertial oscillations are the two dominant frequencies.

Response: Two sets of vertical lines are added in Figure 5 (Figure 6 in the revised version) to clearly show near-inertial (f) and tidal (M2) frequencies. The text in the manuscript is also modified accordingly.

(6) Line 250/251: sentence not understandable. What are the other processes?

Response: Other processes include the background flows, the presence of eddies, variations in sea surface height, non-linear wave-wave and wave-current interactions. This is now mentioned in the manuscript with proper references.

Note: We have improved quality of some figures (Figures 3 and 8 of the original version) without making any changes in the data, scale, and symbols. This is just to improve figures to make them publication quality.

Please also note the supplement to this comment:
https://www.ocean-sci-discuss.net/os-2017-83/os-2017-83-AC1-supplement.pdf

[Figure]

Interactive reply to Anonymous Referee #1 of manuscript OS-2017-83 "Estimation of oceanic sub-surface mixing under a severe cyclonic storm using a coupled atmosphere-ocean-wave model" by Kumar Ravi Prakash et al.

General comments (1) The authors investigate the effects of sub-surface mixing in the ocean under severe storm conditions. The introduction and the reference give the impression that the authors know very well the relevant publication and the overview they give is very nice. To my understanding the novel approach of the article is the use of a coupled atmosphere-ocean-wave model to investigate to simulate the atmospheric and oceanic properties on a very fine scale. The focus is on the generation, propagation and dissipation of kinetic energy in the ocean. I would definitely recommend the publication of the article, although the English is not very well. Almost each sentence is missing an article or the third person "s" is neglected and Ocean is often spelled with capital O. This is not acceptable.

Response: We sincerely thank the Referee for finding it publishable and providing constructive comments that helped to improve the manuscript. We thoroughly checked the manuscript for any missing articles, grammatical mistakes and made necessary corrections. 'Ocean' is now corrected as 'ocean'.

(2) In detail: The abstract is much longer than the conclusions which should be the other way around. And there is not substantial note to the model system used in the article although this is a very important point. Without the model, the investigation could not have happened. So the abstract should focus more on the novel approach and the details of the findings should be discussed in the conclusions.

Response: As suggested by the Referee, we made substantial changes in the Abstract and Conclusions. The abstract is shortened and detailed findings are now discussed in the Conclusions. The model details and model configurations are provided with more details in Section 2 of the revised version. We added a block diagram (Figure 1 in the revised manuscript) to clearly show the exchange of variables between component models WRF, ROMS, and SWAN in the COAWST modeling system. Discussion on variables exchanged added in the manuscript.

(3) Page 3, line 65: NIO is one of the important factors, what are the others?

Response: This sentence is now modified to clear any confusion. The NIO and surface wind stress can generate near-inertial scale mixing at the base of the mixed layer. Other processes such as the nonlinear interaction of NIO and internal tides, and background flows in the ocean can influence the NIO propagation and kinetic energy and affect the mixing process. Effects of other processes are mentioned at appropriate places (sections 3.3.2 and 3.3.3) in the revised manuscript.

(4) Page 8, line 211: where are the 15m to be seen? The link between the description and the figures is not really strong.

Response: We regret this mistake. The mixed layer depth (MLD) was calculated using the density criteria. We have now marked the position of MLD with a thick black line in Figure 4a (Figure 5a in the revised version). Description of figures elaborated to make the link between figures and text, and the flow between sections strong.

**Fig. 1.**

**Supplement:**

[revised manuscript text omitted]

---

## Author Comment (AC2) · 22 Dec 2017

A revised version of the manuscript as per RC1 suggestions is attached as a supplement.

Please also note the supplement to this comment:
https://www.ocean-sci-discuss.net/os-2017-83/os-2017-83-AC2-supplement.pdf

---

## Referee Comment (RC2) · Anonymous Referee #2 · 8 Jan 2018

Comments for the manuscript "Estimation of oceanic sub-surface mixing under a severe cyclonic storm using a coupled atmosphere-ocean-wave model"

General comments

This study presents a case study using a coupled atmosphere-ocean-wave model to investigate the influence of the very severe cyclone storm Phailin on mixing in the upper oceanic layers over the Bay of Bengal. An advantage of coupled model, i.e. interaction and feedback between component models, was utilized and mentioned in the study, however, was not highlighted as it should be. For instance, there is a lack of an analysis for wind speed and wind direction simulated by the atmospheric model

which is an important factor to the mixing and kinetic energy in the ocean during the storm. A case study for one storm event and a time series analysis for only one location in Indian Ocean does not seem to be able to provide robust conclusions. However, the topic is interesting and the introduction provides a good overview about the topic. Therefore, I suggest accepting the paper for publication after major revisions are made.

Major remarks

- Abstract contains too many details which should be moved to the conclusion. Especially in the abstract as well as in the introduction, a clear statement is missing of what is new in this study.

- Configuration (i.e. horizontal resolution, vertical levels, integration time step, etc.) of used models should be provided.

- What is the reason for choosing only one location for time-series analysis? Is there any observation data at this location that can be used to compare with the simulation?

- An evaluation of wind speed and wind direction simulated by WRF is missing although it's relevant for the analysis of D23, MLD, etc.

- The simulated storm track of the stand-alone WRF for this event should be mentioned in the current section 3.2. And how is the performance of stand-alone ROMS in simulating mixing during the storm? One can ask whether the expensive coupled model is really necessary to simulate such event.

- Simulated SST should be analyzed in more details. Although the cooling was captured well during 12-14 Oct but obvious biases occur in 10 and 11 Oct. What is the reason for the biases?

- Conclusion section: for which studies the results of this present study can be applied? Please give examples!

- There is a lack of references at some parts of the manuscript, for instance for the data

sets ECCO2 and ETOPO2 (lines 152, 153) or for the periodogram and Morlet wavelet methods (lines 160, 161). - The English needs strong improvements. I suggest proof reading by a native English speaker.

Minor Comments

- It's not necessary to use the abbreviation VSCS prior to Phailin.

- Line 40-41: what are other important factors to drive the ocean response to the tropical cyclone?

- Lines 72-79: The information does not seem to be important for the present study.

- Citation rule of the journal was not kept in line 95, 98.

- Line 118: The simulation time period should be more specifically described, for example "period of 00 GMT 10 October – 00 GMT 15 October 2013" as "period of 10-15 October 2012" can be understood that the whole day 15 October is also simulated which is not the case in this study.

- Line 164: where is Ef defined?

- Section 3.1 is not a result of this study. It should be moved to the line 81, after the storm Phailin is mentioned.

- Line 271: how to define the rotary spectra of near-inertial wave numbers? Either an equation or a reference is necessary.

- Some sentences are not clear: lines 57-60, 238-242, 247-249. Please rewrite them.

- Line 280-284 should be moved to the conclusion and discussion.

- Too many colors shading steps are used in the figures, please use maximum 15 colors in each figure. For most of the respective panels using half of the number of steps is recommended.

- Caption of figure 1: "is" is missing between "analysis" and "marked".

- Figure 2: It would be good to display the lifetime of the storm (daily should be sufficient) with colors of dates corresponding to the tracks. It will be helpful to see when and where the storm was generated and vanished.

- Caption of figure 2: "Validation" is not necessary here. Can change to: "The Phailin track simulated by the coupled model (black) and the IMD reported track (red)."

- Caption of figure 3: Sequence of upper and lower panels should be switched. For instance: "Daily SST (oC) simulated by the coupled model (upper panel) and observed from the AVHRR satellite (lower panel)."

- Caption of figure 4: "(d)" is missing

- Figure 5: which time period is covered?

- Figure 6: where is the white dashed line?

[Figure]

---

## Referee Comment (RC3) · Anonymous Referee #3 · 9 Jan 2018

Prakash et al. present a study about the effects of very strong storm on the ocean mixing and energy penetration in the water column. Study area is a single station in the Bay of Bengal. A 3D coupled is used for this.

Generally the study is well structured and provides and overview over dominant frequencies found in the models simulated kinetic energy and mixing characteristics. There are some points however, that prevent me from recommending publication in Ocean Science in its present form:

1) It becomes not clear to me what found the frequencies are related to. Are they specific for this special model used? Are they influenced by the frequency of coupling

fluxes between ocean and atmosphere? The coupling time step is also not noted in the manuscript nor which fluxes/variables are exchanged. Or is a physical process possible behind the found frequencies. Could be worth to look at the atm. variable likewise to figure out similar frequencies.

2) The analysis of only one single station is not enough to draw wider robust conclusions. The usage of a 3D high resolution model, however, provides the possibility to extent and confirm the findings for whole domain and eventually further cluster the results according regional characteristics like water depth or hydrographic conditions (background stratification, salinity, river influenced region etc).

3) No motivation for the selection of the abalysis station is given. Is it believed to be representative for a wider region? Why not take one along the main storm track shown in Fig.2 and another one further remote from the storm center. Or just anaylyze a section along the track and another outside the track.?

4) Validation. No attempt is made to demonstrate the ocean modells ability to reproduce generell hydrodynamic and hydrographic characteristics like simply surface SSS and SST. Where are the strengths and weaknesses of this model? When one aims to look at the energy cascade also the winds should be somehow assessed or at least showing the wind characteristics over the station(s) focused.

5) No specific scientific or technical problem is addressed in this study. The manuscript aims at analyzing and quantify subsurface mixing and near inertial mixing. But why is this important and why can this only be done at this single station?

Its difficult to see any broader implication in the manuscript at its present form but rather reads as an execise.

Specific comments

line 27ff Introduction

The motivation of using a coupled atmosphere ocean model should be explained as

this is by no means standard in regional modelling. What is here advantage over stand-alone components which justifies the more computer power required by coupled models. Refer other studies that used coupled models and summarize which improvements or problems they report. What is the added value you expect from a coupled model in you region?

line 76: biological primary productivity is a flux rather not a concentration. I would remove "concentration" in this sentence. Chlorophyll on the other hand is a concentration.

Line 120 – 130.

Here a number very specific parameterization schemes for the atmospheric model are named. Why where these chosen and not others? Are there more options available in this mode? If yes explain why you chosed the ones you explicitly named. Can it be expected that they have impact on the result?

134 – 155: Please tell the reader which variables or fluxes are communicated between the ocean-atmosphere-wave models and which is the coupling time step?

192 – 201

The validation is by far not sufficient here. The track is obviously ok. But no validation for wind is provided which would likewise influence SST and thermocline depth level.

It would be also helpful to judge the general performance of the ocean model for temperature and salinity therby taking account also in situ measurements not only satellite data. Apparently the model exists at least since 2005. Has there ever been a validation undertaken for the ocean model showing general characteristics of the thermocline etc?

What is figure 3 showing? snapshots, daily averages?

Magnitude of cooling: how large is it? Corresponds your single analysis site with the

maximum cooling seen in Fig. 3?

203 – 215

Its no surprise the isotherms are deepening during a storm. What information do you want to give by chosing a 23C isotherm at a single place? Are other processes more important when considering other isotherms?

line 228:

That's intressting. Is it possible to make a more general statement that the baroclinic component is higher than the barotropic here ? how is it in the center of the storm/ in peripherals of the storm?

line 241:

Figure 5 show maximun spectral power in frequency band between roughly 0.03 and 0.04 cycles per hour. Please make more clear which frequencies you attribute to tidal forcings and inertial mixing.

244:

When tidal oscillation were absent at the whole vertical section shown in Fig. 5 how can these oscillation then dominate at the surface (sentence line 240)?

250-251:

Is it meant that NIO dominates the mixing at 14 m depth when windstress would be absent?

252 – 268 (Figure6, Fig. 7):

Here higher order statistics is applied. Which frequency has the input data? The scale in Figure 6 goes until 60 periods/hour, so you need a minimum of 60 seconds in the output frequency of the model, right? Also, Figure 6 show the percentage at 40m depth but the text discusses it at 14 m.

Using filter techniques a number of values for the baroclinic current strengths and kinetic energy are given. The temporal evolution of these parameters is proven to be related to the storm activity. But what else can we learn from that.

Figure 6 contains no white dashed line as mentioned in the text.All this seems not be surprising, so are there more general implications from these singels events analyzed at a single location? This reads more like an exercise.

line 311ff Conclusion

The conclusions lists all the above mentioned phenomena which might be intressting but no attempt is made to put the results in wider context or to draw more general implications for the community. Can corresponding frequencies also be found in atmospheric variables like air temperature, heat fluxes etc? This could be an argument for using coupled models.

No attempt is undertaken to further interpret the found frequencies. Are they model specific i.e. dependent parametrizations, numerical schemes etc or can they be explained physics? And how robust are they when only one anaysis station is used? What is the roles of NIO compared to other processes of vertical mixing especially if you analyze them more on the shelff with shallow water depths?

———————————————

---

## Author Comment (AC3) · 27 Jan 2018

**Responses to the comments received from Anonymous Referee #2 on os-2017-83**

*Our point-by-point responses are given in italic blue font following each comment.*

Comments for the manuscript "Estimation of oceanic sub-surface mixing under a severe cyclonic storm using a coupled atmosphere-ocean-wave model"

General comments
This study presents a case study using a coupled atmosphere-ocean-wave model to investigate the influence of the very severe cyclone storm Phailin on mixing in the upper oceanic layers over the Bay of Bengal. An advantage of coupled model, i.e. interaction and feedback between component models, was utilized and mentioned in the study, however, was not highlighted as it should be. For instance, there is a lack of an analysis for wind speed and wind direction simulated by the atmospheric model which is an important factor to the mixing and kinetic energy in the ocean during the storm. A case study for one storm event and a time series analysis for only one location in Indian Ocean does not seem to be able to provide robust conclusions. However, the topic is interesting and the introduction provides a good overview about the topic. Therefore, I suggest accepting the paper for publication after major revisions are made.

*We thank the anonymous Referee#2 for the constructive comments on the manuscript. As suggested by the Referee, we have now highlighted the advantage of using coupled model in better interaction between atmosphere and ocean and, therefore, better simulation of sea surface temperature and oceanic sub-surface features. A figure showing the validation of atmospheric model simulated wind speed, direction, and surface pressure is now added in the manuscript. A panel showing the wind speed simulated by the atmospheric model is included in the figure where the time series of temperature profile, u- and v- currents, and kinetic energy are shown (Figure 4 of Discussion paper, Figure 6 of Revised paper). As suggested, we have added one more location (on the track) for the time-series analysis in addition to the previously selected off-track location. The figures and text are modified accordingly in the revised manuscript.*

Major remarks
- Abstract contains too many details which should be moved to the conclusion. Especially in the abstract as well as in the introduction, a clear statement is missing of what is new in this study.

*As suggested by the Referee, we have moved the details to the conclusion section. An statement highlighting the novelty of this study is now included in the Abstract (lines 12-14) and Introduction (lines 122-127) in the revised manuscript (with track change).*

- Configuration (i.e. horizontal resolution, vertical levels, integration time step, etc.) of used models should be provided.

*The details of model configuration including horizontal resolution, vertical levels, integration time step, etc. are now provided in the revised manuscript (Lines 204-213).*

- What is the reason for choosing only one location for time-series analysis? Is there any observation data at this location that can be used to compare with the simulation?

*We have now added another location on the track of cyclone and compared results with the existing off-track location. However, the larger kinetic energy and mixing were found at the off-track location as compared to the on-track location. There was no observational data at the off-track location but the selection was based on the maximum surface cooling observed*

*at this location. The revised figures to show analysis at both the locations are included in the revised manuscript.*
- An evaluation of wind speed and wind direction simulated by WRF is missing although it's relevant for the analysis of D23, MLD, etc.

*We agree with the Referee suggestion. A new figure (Figure 4 in revised manuscript) added to show the validation of the WRF simulated wind speed and direction with the buoy BD09 measurements.*

- The simulated storm track of the stand-alone WRF for this event should be mentioned in the current section 3.2. And how is the performance of stand-alone ROMS in simulating mixing during the storm? One can ask whether the expensive coupled model is really necessary to simulate such event.

*The stand-alone WRF model was found to simulate Phailin track almost similar (figure shown below but not included in the paper) to the WRF in coupled configuration. However, the intensity (surface wind speed) in WRF stand-alone model was higher as compared to the coupled model (Figure 4 of the revised manuscript). The WRF in coupled model configuration shows better performance in simulating the surface wind speed and pressure during Phailin. The exchange of wave parameters with the WRF model in coupled configuration provides realistic sea surface roughness that resulted in improvement of surface wind speed (included in section 3.1 of the revised manuscript).*

[Figure]

*Figure: Comparison of Phailin tracks simulated by stand-alone WRF model and coupled model with the IMD reported track.*

*The stand-alone ROMS model forced with the WRF winds in un-coupled mode overestimates the cyclone-induced cooling with -2.2 ºC bias in SST on 13-14 October. The stronger surface winds in stand-alone WRF cause the larger cold bias in stand-alone ROMS model. The SST comparison figure now includes stand-alone ROMS model SST as well (Figure 5 of the revised paper).*

- Simulated SST should be analyzed in more details. Although the cooling was captured well during 12-14 Oct but obvious biases occur in 10 and 11 Oct. What is the reason for the biases?

*The coupled model captures the SST spatial pattern reasonably well with about -0.5ºC bias in northwestern BoB on 13-14 October. This order of bias in SST could be resulted from the errors in initial and boundary conditions provided to the model. The biases on initial days of*

*10-11 October are due to the biases in ECCO2 data used to initialize the model. These points are now mentioned in lines 313-315 of the revised manuscript.*

- Conclusion section: for which studies the results of this present study can be applied? Please give examples!

*The coupled model found to be a useful tool to investigate air-sea interaction, kinetic energy propagation, and mixing in the upper-ocean. The proper representation of kinetic energy propagation and oceanic mixing have applications in improving the intensity prediction of cyclone, storm surge forecasting, and biological productivity. These points are included in the conclusion section of the revised manuscript.*

- There is a lack of references at some parts of the manuscript, for instance for the data sets ECCO2 and ETOPO2 (lines 152, 153) or for the periodogram and Morlet wavelet methods (lines 160, 161).

*References are now added for ECCO2, ETOPO2, periodogram, and Morlet wavelet methods.*

- The English needs strong improvements. I suggest proof reading by a native English speaker.

*We have thoroughly checked the manuscript for any English language or grammatical errors and corrected the same.*

Minor Comments
- It's not necessary to use the abbreviation VSCS prior to Phailin.
*The abbreviation 'VSCS' prior to Phailin has been removed.*

- Line 40-41: what are other important factors to drive the ocean response to the tropical cyclone?
*The sentence has been modified to make it clear (in lines 53-54 of the revised paper).*

- Lines 72-79: The information does not seem to be important for the present study.
*These lines are deleted.*

- Citation rule of the journal was not kept in line 95, 98.
*Corrected.*

- Line 118: The simulation time period should be more specifically described, for example "period of 00 GMT 10 October – 00 GMT 15 October 2013" as "period of 10-15 October 2012" can be understood that the whole day 15 October is also simulated which is not the case in this study.

*Corrected as suggested by the Referee.*

- Line 164: where is Ef defined?
*Ef denotes the inertial baroclinic kinetic energy, defined in line 261 of the revised manuscript.*

- Section 3.1 is not a result of this study. It should be moved to the line 81, after the storm Phailin is mentioned.

*Agreed, the write-up is now moved to lines 110-119 in Introduction (revised manuscript).*

- Line 271: how to define the rotary spectra of near-inertial wave numbers? Either an

equation or a reference is necessary.

*The wave-number rotary spectra provides a clear picture of wind energy distribution in the sub-surface water, which is used in the present study for the near-inertial oscillations. References are now added to the rotary spectra in line 448 in the revised manuscript.*

- Some sentences are not clear: lines 57-60, 238-242, 247-249. Please rewrite them.
*Rewritten the sentences to make them clear in lines 79-83, 392-397, 406-410, respectively.*

- Line 280-284 should be moved to the conclusion and discussion.
*The lines are moved (with suitable modifications) to the conclusion section.*

- Too many colors shading steps are used in the figures, please use maximum 15 colors in each figure. For most of the respective panels using half of the number of steps is recommended.
*Corrected as suggested by the Referee.*

- Caption of figure 1: "is" is missing between "analysis" and "marked".

*Caption of Figure 1 (Figure 2 in revised version) is corrected.*

- Figure 2: It would be good to display the lifetime of the storm (daily should be sufficient) with colors of dates corresponding to the tracks. It will be helpful to see when and where the storm was generated and vanished.
*The lifetime of the storm is now indicated in the figure (Figure 3 in the revised version).*

- Caption of figure 2: "Validation" is not necessary here. Can change to: "The Phailin track simulated by the coupled model (black) and the IMD reported track (red)."
*Corrected. (Figure 3 in the revised version)*

- Caption of figure 3: Sequence of upper and lower panels should be switched. For instance: "Daily SST (oC) simulated by the coupled model (upper panel) and observed from the AVHRR satellite (lower panel)."
*Corrected as suggested. (Figure 5 in the revised version).*

- Caption of figure 4: "(d)" is missing
*Corrected. (Figure 6 in the revised version).*

- Figure 5: which time period is covered?
*Figure caption is modified to make it clear. The power spectrum analysis was performed on the simulation period (10-14 October). The frequency ranges of near-inertial oscillations (f) and semidiurnal tidal constituent (M2) are shown with the vertical lines in the figure. (Figure 7 in the revised version).*

- Figure 6: where is the white dashed line?

*There was no white dashed line in the figure. Now the caption is corrected. (Figure 9 in the revised version).*

---

## Author Comment (AC4) · 27 Jan 2018

**Responses to the comments received from Anonymous Referee #3 on os-2017-83**

*We thank the anonymous Referee#3 for the constructive comments on the manuscript. Our point-by-point responses are given in italic blue font following each comment.*

Prakash et al. present a study about the effects of very strong storm on the ocean mixing and energy penetration in the water column. Study area is a single station in the Bay of Bengal. A 3D coupled is used for this.

Generally the study is well structured and provides and overview over dominant frequencies found in the models simulated kinetic energy and mixing characteristics. There are some points however, that prevent me from recommending publication in Ocean Science in its present form:

1) It becomes not clear to me what found the frequencies are related to. Are they specific for this special model used? Are they influenced by the frequency of coupling fluxes between ocean and atmosphere? The coupling time step is also not noted in the manuscript nor which fluxes/variables are exchanged. Or is a physical process possible behind the found frequencies. Could be worth to look at the atm. variable likewise to figure out similar frequencies.

*The near-inertial frequencies are process and location (latitude) depended and not on the selection of model, model configuration or frequency of coupling fluxes between ocean and atmosphere. The frequency ranges of near-intertial oscillations (f) at the selected location and semidiurnal tidal constituent (M2) are now marked with two pairs of vertical lines in Figure 7 of the revised manuscript. The high energy of near-inertial oscillations (NIO) found in our study is generated due to the strong cyclonic winds. As suggested by the Referee, we have performed the power spectrum analysis on air-temperature and found similar frequencies. The figure is given below.*

[Figure]

*Figure: The power spectrum analysis ($m^2 s^{-1}$) performed on the simulation period at the on-track (left panel) and off-track (right panel) locations.*

*In the revised manuscript, we have added an on-track location in addition to the existing off-track location (the two locations are shown in Figure 2 of the revised version) and results were compared for the kinetic energy. Similar NIO frequencies were found in air-temperature as that of oceanic currents. Further, the strength of NIO frequencies are stronger at the off-track location as compared to the on-track location both in the atmosphere and ocean. The power associated with NIO in the atmosphere (air-temperature) is about an order weaker than the oceanic NIO. As the aim of this paper is to estimate the kinetic energy propagation and mixing within the ocean, we have not attempted to analyse the energy distribution in the atmospheric column and, therefore, the air-temperature power spectrum figure is not being added to the manuscript.*

*The coupling time step (600 s) are now mentioned in lines 244-245 of the revised version. The fluxes/variables exchanged between the component atmosphere/ocean/wave models are now clearly mentioned in lines 169-181 of the revised version. A new figure (Figure 1) is included to show the variables exchanged between the model components.*

2) The analysis of only one single station is not enough to draw wider robust conclusions. The usage of a 3D high resolution model, however, provides the possibility to extent and confirm the findings for whole domain and eventually further cluster the results according regional characteristics like water depth or hydrographic conditions (background stratification, salinity, river influenced region etc).

*In the revised manuscript, we have added an on-track location in addition to the existing off-track location (the two locations are shown in Figure 2 of the revised version) and results were compared (in Figures 6, 7, and 8 in the revised version). The analysis showed weaker NIO at the on-track location as compared to the off-track location. We have limited our analysis to the region influenced by the strong cyclonic winds that transfer the kinetic energy to the oceanic column.*

*A few studies have shown that the near-inertial energy rapidly decreases in the shallower depths towards the coast. The spatial distribution of near-inertial energy is primarily controlled by the boundary effect for inertial oscillations (Chen et al., 2017). The NIO energy found to decline with the decreasing depth and vanish in the coastal regions (Schahinger, 1988; Chen et al., 2017). Our 3-d coupled model simulations are performed for the period of a very severe cyclone Phailin. In the presence of strong cyclonic winds, impact of the differences in salinity (river influenced) on NIO as compared to a location under the storm would be difficult to access. Such a study would require a normal weather condition with uniform winds (idealistic winds) over the whole domain. However, this could be an interesting problem to be explored in the salt-stratified BoB in a future study.*

3) No motivation for the selection of the abalysis station is given. Is it believed to be representative for a wider region? Why not take one along the main storm track shown in Fig.2 and another one further remote from the storm center. Or just anaylyze a section along the track and another outside the track.?

*As suggested by the Referee, we have now added another location on the track of cyclone and compared results with the existing off-track location (the two locations are shown in Figure 2 of the revised version). However, the larger kinetic energy and mixing were found at the off-track location as compared to the on-track location. The selection of the off-track location was based on the maximum surface cooling observed at this location indicating strong mixing. The revised figures (Figures 6, 7, and 8 in the revised version) to show analysis at both the locations are included in the revised manuscript.*

4) Validation. No attempt is made to demonstrate the ocean modells ability to reproduce generell hydrodynamic and hydrographic characteristics like simply surface SSS and SST. Where are the strengths and weaknesses of this model? When one aims to look at the energy cascade also the winds should be somehow assessed or at least showing the wind characteristics over the station(s) focused.

*The performance of the coupled atmosphere-ocean model in simulating the oceanic parameters temperature, salinity, and currents during the Phailin is accessed in Prakash and Pant (2017). The strength of the coupled model is in simulating better oceanic and atmospheric features as compared to the stand-alone oceanic/atmospheric model. For example, the stand-alone WRF overestimates surface wind speed (comparison shown between coupled and stand-alone WRF against the buoy measurements in Figure 4 of the revised version). The stand-alone ocean model ROMS produces cold bias >-2.0 deg C (comparison shown in Figure 5 of the revised version). The better simulation of SST in coupled configuration resulted from the improvement in the wind speed and heat-fluxes.*

*As suggested, we have now included a panel showing the wind speed (in Figure 6 of the revised version) along with the profiles of temperature, u- and v-current, and barotropic/ baroclinic kinetic energy at both on-track and off-track locations.*

5) No specific scientific or technical problem is addressed in this study. The manuscript aims at analyzing and quantify subsurface mixing and near inertial mixing. But why is this important and why can this only be done at this single station? Its difficult to see any broader implication in the manuscript at its present form but rather reads as an execise.

*The study is first of its kind in the Bay of Bengal utilizing a 3-D coupled atmosphere-ocean-wave model to estimate the oceanic sub-surface mixing in the presence of near-inertial oscillations during a severe cyclone (mentioned in lines 12-14, 123-127 in the revised version). The study has important practical/societal implications. The proper representation of kinetic energy propagation and oceanic mixing have applications in improving the intensity prediction of cyclone, storm surge forecasting, and biological productivity (mentioned in lines 525-529 in the revised version). The analysis is now performed at two locations i.e. on-track and off-track, as suggested by the Referee.*

Specific comments
line 27ff Introduction
The motivation of using a coupled atmosphere ocean model should be explained as this is by no means standard in regional modelling. What is here advantage over stand-alone components which justifies the more computer power required by coupled models. Refer other studies that used coupled models and summarize which improvements or problems they report. What is the added value you expect from a coupled model in you region?

*A number of studies have used a coupled atmosphere-ocean model over a regional domain to address a variety of atmospheric/oceanic processes (Zambon et al., 2014; Warner et al., 2010). The regional coupled model is particularly useful in simulating atmospheric/oceanic conditions in storm conditions. There are several other regional studies (Warner et al., 2010, Ricchi et al., 2016, Nelson et al., 2014, Wu et al., 2016) for different storm cases and extreme events by utilizing the coupled atmosphere-ocean-wave model.*

*The coupled atmosphere-ocean model found to improve the intensity of cyclonic storm when compared to the uncoupled model over different oceanic regions (Warner et al., 2010; Zambon et al., 2014; Srinivas et al., 2016; Wu et al., 2016). Zambon et al. (2014) compared*

*the simulations from the coupled atmosphere-ocean and uncoupled models and reported significant improvement in the intensity of storm in the coupled case as compared to the uncoupled case. The uncoupled atmospheric model produced large ocean-atmosphere enthalpy fluxes and stronger winds in the cyclone (Srinivas et al., 2016). When the atmospheric model WRF was allowed interactions with the ocean model, the SST found to be more realistic as compared to warm bias in the stand-alone WRF (Warner et al., 2010). Wu et al. (2016) demonstrated the advantage of using a coupled model over the uncoupled model in better simulation of typhoon Megi's intensity. These points are now included in lines 57-67 of the revised version.*

*In our region, there are marked improvements in the wind speed in the atmospheric model and SST in the oceanic model component in the coupled configuration as compared to the stand-alone atmospheric model and stand-alone ocean model (Figures 4 and 5 in the revised version). The mean sea level pressure was lower and surface wind speed was higher in the stand-alone model as compared to the coupled model, which was close to the buoy (BD09) measurements (Figure 4 in revised version). The overestimated wind speed in the stand-alone atmospheric model give rise to the larger heat loss by ocean in the stand-alone ocean model resulting into the cold SST bias (Figure 5 in revised version). The improvement in the surface wind speed in coupled configuration is due to the better representation of sea surface roughness and heat-fluxes.*

line 76: biological primary productivity is a flux rather not a concentration. I would remove "concentration" in this sentence. Chlorophyll on the other hand is a concentration.

*This paragraph on the biological productivity has been deleted in the revised manuscript as per suggestions from another Referee.*

Line 120 – 130.
Here a number very specific parameterization schemes for the atmospheric model are named. Why where these chosen and not others? Are there more options available in this mode? If yes explain why you chosed the ones you explicitly named. Can it be expected that they have impact on the result?

*The specific parameterization schemes used in this study are based on the sensitivity experiments of the parameterization schemes in the same atmospheric model WRF by ourselves and others (Osuri et al., 2012). There are other parameterization schemes available in the model but over our region, the selected schemes performed better. Similar parameterization schemes were used in the coupled atmosphere-ocean model by Glenn et al. (2016), Zambon et al. (2014), Warner et al. (2010).*

134 – 155: Please tell the reader which variables or fluxes are communicated between the ocean-atmosphere-wave models and which is the coupling time step?

*The coupling time step (600 s) are now mentioned in lines 244-245 of the revised version. The fluxes/variables exchanged between the component atmosphere/ocean/wave models are now clearly mentioned in lines 169-181 of the revised version. A new figure (Figure 1) is included to show the variables exchanged between the model components.*

192 – 201
The validation is by far not sufficient here. The track is obviously ok. But no validation for wind is provided which would likewise influence SST and thermocline depth level. It would be also helpful to judge the general performance of the ocean model for temperature and salinity therby taking account also in situ measurements not only satellite data. Apparently

the model exists at least since 2005. Has there ever been a validation undertaken for the ocean model showing general characteristics of the thermocline etc?

*A validation of wind at the buoy BD09 location is now provided in Figure 4 of the revised version. For comparison, the stand-alone atmospheric model and coupled model simulated winds are plotted together with the buoy measurements. The general performance of the ocean model in coupled configuration for the Phailin cyclone has been thoroughly discussed in our published article (Prakash and Pant, 2017). The similar model has been found capable in simulating realistic oceanic features in various storm conditions (Warner et al., 2010, Nelson et al., 2014, Wu et al., 2016, Zambon et al., 2014).*

What is figure 3 showing? snapshots, daily averages?
Magnitude of cooling: how large is it? Corresponds your single analysis site with the maximum cooling seen in Fig. 3?
*The SST values shown in Figure 3 (Figure 5 in revised version) are daily averages. Figure caption is modified to reflect this. In the daily-average plot (Figure 3 of old version), the magnitude of cooling was up to -2.5 °C but in the hourly time series plot (Figure 4a in old version; Figure 6 in revised version) up to -3.5°C maximum cooling was observed.*

203 – 215
Its no surprise the isotherms are deepening during a storm. What information do you want to give by chosing a 23C isotherm at a single place? Are other processes more important when considering other isotherms?

*The thermocline, defined as the depth of maximum temperature gradient, is usually referred to a location dependent isotherm depth (Kessler, 1990; Wang et al, 2000). Over the BoB region, the depth of 23ºC isotherm (D23) found to be an appropriate representative depth of the thermocline (Girishkumar et al., 2013). These points are now mentioned in lines 339-342 of the revised version.*

line 228:
That's intressting. Is it possible to make a more general statement that the baroclinic component is higher than the barotropic here ? how is it in the center of the storm/ in peripherals of the storm?

*After performing the analysis over the on-track location (centre of the storm) in addition to the existing off-track location, we can infer that the kinetic energy associated with the baroclinic component found to be much higher than the barotropic component of current in the centre as well as off-track location in the Phailin (Figure 6 of the revised version). Please notice that barotropic currents plotted with a multiplication factor of $10^2$ to plot both baroclinic and barotropic current components in the same scale (this multiplication factor is mentioned in figure caption).*

line 241:
Figure 5 show maximun spectral power in frequency band between roughly 0.03 and 0.04 cycles per hour. Please make more clear which frequencies you attribute to tidal forcings and inertial mixing.

*The frequencies attributed to the near-inertial oscillations and tidal forcing are now clearly marked with f and M2, respectively in Figure 7 of the revise version.*

244:
When tidal oscillation were absent at the whole vertical section shown in Fig. 5 how can these oscillation then dominate at the surface (sentence line 240)?

*The sentence has been elaborated and modified to make it clear (lines 400-406 in the revised version).*

250-251:
Is it meant that NIO dominates the mixing at 14 m depth when windstress would be absent?
*We meant that NIO dominates the mixing at 14 m depth in presence of local wind stress that dominated the mixing compared to any other process. Other processes include the background flows, the presence of eddies, variations in sea surface height, non-linear wave-wave and wave-current interactions (Guan et al., 2014; Park and Watts, 2005). The sentence has been modified in lines 411-415 in the revised version.*

252 – 268 (Figure6, Fig. 7):
Here higher order statistics is applied. Which frequency has the input data? The scale in Figure 6 goes until 60 periods/hour, so you need a minimum of 60 seconds in the output frequency of the model, right? Also, Figure 6 show the percentage at 40m depth but the text discusses it at 14 m.
*We regret this mistake in the unit. We have now corrected the unit as Period (days) in Figure 9 of the revised version. The frequency of the input data is 60 min. The typo mistake of 40 m is corrected as 14 m in figure caption.*

Using filter techniques a number of values for the baroclinic current strengths and kinetic energy are given. The temporal evolution of these parameters is proven to be related to the storm activity. But what else can we learn from that.

*The filter techniques used to estimate the strength of near-inertial oscillations (NIO) in the frequency range of 0.028 to 0.038 cycles h-1 at the selected locations. The analysis was helpful to understand the downward propagation of kinetic energy and decay of NIO with the increasing depth. We realized that profiles of filteres baroclinic meridional (Vf) and zonal (Uf) currents based on the filter technique was not adding any new information to the paper. Therefore, we have now replaced the Figure 7 (old version) with Figure 8 (in revised version). The figure description has been modified accordingly in lines 416-428 in revised version.*

Figure 6 contains no white dashed line as mentioned in the text. All this seems not be surprising, so are there more general implications from these singels events analyzed at a single location? This reads more like an exercise.
*There was no white dashed line in the figure and its mention in the text and figure caption has been deleted. The analysis is now performed at two locations- one at the on-track and another at the off-track location.*

line 311ff Conclusion
The conclusions lists all the above mentioned phenomena which might be intressting but no attempt is made to put the results in wider context or to draw more general implications for the community. Can corresponding frequencies also be found in atmospheric variables like air temperature, heat fluxes etc? This could be an argument for using coupled models.

*The study has important practical/societal implications. The proper representation of kinetic energy propagation and oceanic mixing have applications in improving the intensity prediction of cyclone, storm surge forecasting, and biological productivity (mentioned in lines 525-529 in the revised version). As suggested by the Referee, we have performed the power spectrum analysis on air-temperature and found similar frequencies. The figure is given above with its description.*

No attempt is undertaken to further interpret the found frequencies. Are they model specific i.e. dependent parametrizations, numerical schemes etc or can they be explained physics? And how robust are they when only one anaysis station is used? What is the roles of NIO compared to other processes of vertical mixing especially if you analyze them more on the shelff with shallow water depths?

*The near-inertial frequencies are process and location (latitude) depended and not on the selection of model, model configuration, parameterization, or numerical schemes used in the coupled model. The high energy of near-inertial oscillations (NIO) found in our study is generated due to the strong cyclonic winds. Now, the analysis has been performed at two locations- on-track and off-track in the cyclone Phailin. Through the filter technique analysis, we have shown (in Figures 7 and 8 in the revised version) that NIO were the dominant frequency signals in presence of cyclone-induced local wind stress that dominated the vertical mixing in our study period during a cyclone. Our analysis was limited to the locations under the influence of strong cyclonic winds that would lead to strong mixing and associated cooling. However, a few studies have shown that the near-inertial energy rapidly decreases in the shallower depths towards the coast. The spatial distribution of near-inertial energy is primarily controlled by the boundary effect for inertial oscillations (Chen et al., 2017). The NIO energy found to decline with the decreasing depth and vanish in the coastal regions (Schahinger, 1988; Chen et al., 2017), mentioned in lines 96-99 of the revised version.*

*References:*

*Chen, S. and Chen, D. and Xing, J.: A study on some basic features of inertial oscillations and near-inertial internal waves, Ocean Science Vol. 13 (5), pp 829-836, 2017.*

*Girishkumar, M. S., Ravichandran, M., Han, W.: Observed intraseasonal thermocline variability in the Bay of Bengal. J. Geophys. Res. Oceans, 118, 3336–3349, doi:10.1002/jgrc.20245, 2013.*

*Glenn, S., and Coauthors: Stratified coastal ocean interactions with tropical cyclones. Nat. Commun., 7, 10887, doi:10.1038/ncomms10887, 2016.*

*Guan, S., Zhao, W., Huthnance, J. Tian, J., and Wang, J.: Observed upper ocean response to typhoon Megi (2010) in the Northern South China Sea. J. Geophys. Res. Oceans, 119, 3134– 3157, doi:10.1002/2013JC009661, 2014.*

*Kessler, W. S.: Observations of long Rossby waves in the northern tropical Pacific. J. Geophys. Res., 95, 5183–5217, 1990.*

*Osuri K. K. U. C. Mohanty, A. Routray, M. A. Kulkarni, M. Mohapatra: Customization of WRF-ARW model with physical parameterization schemes for the simulation of tropical cyclones over North Indian Ocean, Nat Hazards, 63:1337–1359, DOI 10.1007/s11069-011-9862-0, 2012.*

*Park, J.H., and Watts, D. R.: Near-inertial oscillations interacting with mesoscale circulation in the southwestern Japan/East Sea. Geophys. Res. Lett., 32, L10611, doi: 10.1029/2005GL022936, 2005.*

*Prakash K.R., Vimlesh Pant: Upper oceanic response to tropical cyclone Phailin in the Bay of Bengal using a coupled atmosphere-ocean model, Ocean Dynamics, 67, 51-64, doi:10.1007/s10236-016-1020-5, 2017.*

*Schahinger, R.B.: Near inertial motion on the south Australian shelf. J. Phys. Oceanogr., 18(3), 492-504, 1988.*

*Srinivas, C. V., Mohan,G. M., Naidu, C. V., Baskaran, R., Venkatraman B. :Impact of air-sea coupling on the simulation of tropical cyclones in the North Indian Ocean using a simple 3-D ocean model coupled to ARW, J. Geophys. Res. Atmos., 121, 9400,9421, doi:10.1002/2015JD024431, 2016.*

*Nelson J., He R., Warner J.C., Bane J.: Air-sea interactions during strong winter extratropical storms. Ocean Dyn. 64(9), 1233-1246, doi:10.1007/s10236-014-0745-2, 2014.*

*Ricchi, A., Miglietta, M.M., Falco, P.P., Benetazzo, A., Bonaldo, D., Bergamasco, A., Sclavo, M., Carniel, S.: On the use of a coupled ocean-atmosphere-wave model during an extreme Cold Air Outbreak over the Adriatic Sea. Atmos. Res. 172-173, 48–65, doi:10.1016/j.atmosres.2015.12.023, 2016.*

*Zambon J.B., He R., Warner J.C.: Investigation of hurricane Ivan using the coupled ocean–atmosphere–wave–sediment transport (COAWST) model. Ocean Dyn. 64(11),1535–1554, 2014.*

*Wang, B., Wu, R., and Lukas R.: Annual adjustment of the thermocline in the tropical Pacific Ocean, J. Clim., 13, 596–616, 2000.*

*Warner J.C., Armstrong B., He R., Zambon J.B.: Development of a coupled ocean–atmosphere–wave–sediment transport (COAWST) modeling system. Ocean Model 35,230–244, doi:10.1016/j. oceanmod.2010.07.010, 2010.*

*Wu, C.-C., W.-T. Tu, I.-F. Pun, I-I. Lin, and M. S. Peng. Tropical cyclone-ocean interaction in Typhoon Megi (2010)—A synergy study based on ITOP observations and atmosphere-ocean coupled model simulations, J. Geophys. Res. Atmos., 121, 153–167, doi:10.1002/2015JD024198, 2016.*

---

## Author Comment (AC6) · 27 Jan 2018

Revised Manuscript with track changes attached.

Please also note the supplement to this comment:
https://www.ocean-sci-discuss.net/os-2017-83/os-2017-83-AC6-supplement.pdf
* * *

---

## Referee Report (RR1)

**Comments on the paper „**Estimation of oceanic sub-surface mixing under a severe cyclonic storm using a coupled atmosphere-ocean-wave model**"**

**General comments**

The authors investigate the effects of sub-surface mixing in the ocean under severe storm conditions. The introduction and the reference give the impression that the authors know very well the relevant publication and the overview they give is very nice. To my understanding the novel approach of the article is the use of a coupled atmosphere-ocean-wave model to investigate to simulate the atmospheric and oceanic properties on a very fine scale. The focus is on the generation, propagation and dissipation of kinetic energy in the ocean.

The review improved the paper a lot although some things still have to be corrected.

In detail:

Please avoid capital letters when it is not a special name. And it would be nicer if a native speaker would have checked the language.

l 76: kinetic energy

l 80: The NIO is found to decline

l94: "discussed" is better than "accessed".

L 148:  Phailin

L 164: grid-scale

L170: "starching" parameter. This is a major error. There is no such thing as a starching parameter. Please read the model manual again and correct this word!

l 239: stand-alone and coupled WRF

l241: simulated a larger pressure drop

Figure 4: Did you discuss the high frequency variability which can be seen in the figures somewhere? Why are both model configurations not able to reproduce this feature?

L 246ff: Please mention first what the top level boundary conditions in the stand alone ROMS simulation are, so it is nicer for the reader to understand what is going on. And what about a reanalysis driven simulation? Might be that the SST would also be represented well by the ocean model then.

L 410: of the severe cyclonic storm

Fig 7: hard to see the lines against the dark colours.

---

## Author Response (AR2)

**Responses to the comments received on manuscript OS-2017-83**

**Responses to comments from Anonymous Referee #1**

The authors investigate the effects of sub-surface mixing in the ocean under severe storm conditions. The introduction and the reference give the impression that the authors know very well the relevant publication and the overview they give is very nice. To my understanding the novel approach of the article is the use of a coupled atmosphere-ocean-wave model to investigate to simulate the atmospheric and oceanic properties on a very fine scale. The focus is on the generation, propagation and dissipation of kinetic energy in the ocean.
The review improved the paper a lot although some things still have to be corrected.

In detail:
Please avoid capital letters when it is not a special name. And it would be nicer if a native speaker would have checked the language.

*Response:*
*We sincerely thank the anonymous Referee for his suggestions. We thoroughly checked the manuscript for any grammatical errors and corrected the same.*

l 76: kinetic energy
*Corrected*

l 80: The NIO is found to decline
*Corrected*

l94: "discussed" is better than "accessed".
*Corrected*

L 148: the Phailin
*Corrected*

L 164: grid-scale
*Corrected*

L170: "starching" parameter. This is a major error. There is no such thing as a starching parameter. Please read the model manual again and correct this word!

*It was a typo. We replaced the word 'starching' with 'stretching'.*

l 239: stand-alone and coupled WRF
*Corrected*

l241: simulated a larger pressure drop
*Corrected.*

Figure 4: Did you discuss the high frequency variability which can be seen in the figures somewhere? Why are both model configurations not able to reproduce this feature?

*Response: The high frequency variability in mean sea level pressure (MSLP) is primarily due to radiational effects (explained by Pugh, 1987) which are noticed in the buoy measured data. The models are not able to capture this radiation-dependent high frequency variability*

*in the mean MSLP over the cyclone-influenced region. However, the pressure drop associated with the passage of cyclone (during 10-12 October) was well simulated by the model, particularly by the coupled model. This description is now added in lines 246-249 in the revised version.*

L 246ff: Please mention first what the top level boundary conditions in the stand alone ROMS simulation are, so it is nicer for the reader to understand what is going on. And what about a reanalysis driven simulation? Might be that the SST would also be represented well by the ocean model then.

*Response: The following sentence is added in lines 256-257 of the revised version. 'The stand-alone WRF simulated parameters were used to provide surface boundary conditions in the stand-alone ROMS model.'*

*For a trial, we performed the cyclone Phailin simulations with a reanalysis data and found large bias in SST and currents in the ocean. The intensity and tracks of strong cyclones are not properly represented in the reanalysis data and, therefore, the reanalysis driven simulations are not expected to provide better results than a dynamically coupled atmosphere-ocean model.*

L 410: of the severe cyclonic storm
*Corrected*

Fig 7: hard to see the lines against the dark colours.
*The lines are now made thicker to make these clearly visible in Figure 7.*

*Reference:*

*Pugh, D.T.: Tides, Surges and Mean Sea-Level, John Wiley & Sons, Chichester, 472 pp., 1987.*

**Responses to comments from Reviewer #3**

Comment:
Figures 4 (atmosphere validation) 5 (SST) demonstrate so obviously the better performance of the coupled model over the uncoupled model that one could think of mentioning this more prominent (maybe also in the abstract). Especially the improvement in SST and extreme weather phenomena has also found in coupled modeling studies for totally different regions (e.g Europe, Baltic, Jeworek et al., 2017, Hagemann, et al., 2017, Gröger et al., 2015). However, I understand that the added value of coupling is not the focus of the present study. On the other hand this could make the study more interesting to research communities outside the Indian ocean and would likely increase the potential for referencing in future oceanoraphic literature. The publication is however acceptable as it is now.

Ho-Hagemann, H.T.M., Gröger, M., Rockel, B., Zahn, M., Geyer, B., Meier, H.E.M, 2017, Effects of air-sea coupling over the North Sea and the Baltic Sea on simulated summer precipitation over Central Europe, Clim Dyn,49: 3851. https://doi.org/10.1007/s00382-017-3546-8

Jeworrek, J., Wu, L., Dieterich, C., and Rutgersson, A., 2017: Characteristics of convective snow bands along the Swedish east coast, Earth Syst. Dynam., 8, 163-175, https://doi.org/10.5194/esd-8-163-2017.

Gröger M, Dieterich C, Meier HEM, Schimanke S (2015) Thermal air-sea coupling in hindcast simulations for the North Sea and Baltic Sea on the NW European shelf. Tellus A Dyn Meteorol Oceanogr 67(1):26911. doi: 10.3402/tellusa.v67.26911

*Response:*
*We sincerely thank the anonymous Referee for this suggestion. We have added following sentence in the Abstract,*
*'The coupled model found to improve the sea surface temperature over the uncoupled model.' In lines 9-10 of the revised version.*

*The suggested references are added in line 55 of the revised version and included in the References list.*